# Analysis of long and short enhancers in melanoma cell states

David Mauduit[1,2], Ibrahim Ihsan Taskiran[1,2], Liesbeth Minnoye[1,2], Maxime de Waegeneer[1,2], Valerie Christiaens[1,2], Gert Hulselmans[1,2], Jonas Demeulemeester[1,2,3], Jasper Wouters[1,2], Stein Aerts[1,2]*

[1]VIB-KU Leuven Center for Brain & Disease Research, Leuven, Belgium; [2]KU Leuven, Department of Human Genetics KU Leuven, Leuven, Belgium; [3]Cancer Genomics Laboratory, The Francis Crick Institute, London, United Kingdom

**Abstract** Understanding how enhancers drive cell-type specificity and efficiently identifying them is essential for the development of innovative therapeutic strategies. In melanoma, the melanocytic (MEL) and the mesenchymal-like (MES) states present themselves with different responses to therapy, making the identification of specific enhancers highly relevant. Using massively parallel reporter assays (MPRAs) in a panel of patient-derived melanoma lines (MM lines), we set to identify and decipher melanoma enhancers by first focusing on regions with state-specific H3K27 acetylation close to differentially expressed genes. An in-depth evaluation of those regions was then pursued by investigating the activity of overlapping ATAC-seq peaks along with a full tiling of the acetylated regions with 190 bp sequences. Activity was observed in more than 60% of the selected regions, and we were able to precisely locate the active enhancers within ATAC-seq peaks. Comparison of sequence content with activity, using the deep learning model DeepMEL2, revealed that AP-1 alone is responsible for the MES enhancer activity. In contrast, SOX10 and MITF both influence MEL enhancer function with SOX10 being required to achieve high levels of activity. Overall, our MPRAs shed light on the relationship between long and short sequences in terms of their sequence content, enhancer activity, and specificity across melanoma cell states.

*For correspondence:
stein.aerts@kuleuven.be

Competing interest: The authors declare that no competing interests exist.

## Editor's evaluation

This study describes an integrative analysis of the location, regulation, and function of melanoma cell state-specific enhancer elements. By comparing enhancer activity through massively parallel reporter assays, chromatin features, and underlying TF binding profiles in melanocytic and mesenchymal-like melanoma cell states, the authors identify candidate regulators and mechanisms that explain enhancer activity and specificity in melanoma biology. These findings will be of broad interest to those seeking to understand cell type- or cell identify-specific gene regulation at the level of transcriptional and epigenetic control of *cis*-regulatory elements.

## Introduction

Enhancers are crucial regulatory regions in the genome that control cell type-specific gene expression. Identifying enhancers helps to better understand cell identity and is key to develop therapies targeting a singular relevant cell type in a disease. To date, accurate prediction of enhancer location and cell-type activity remains a challenge. Both the presence and clustering of transcription factor binding sites (TFBSs) are good predictors of enhancer activity (*Gasperini et al., 2020*; *King et al., 2020*). Yet, such an approach requires prior knowledge of the *cis*-regulatory grammar in the studied cell types as only a small proportion of the TFBSs found in the genome are bound by the corresponding transcription

factor (TF) (*Yáñez-Cuna et al., 2012*). Another strategy to identify candidate enhancers is to use active enhancer marks such as H3K27ac and chromatin accessibility (*Gray et al., 2017*; *Minnoye et al., 2021*; *Rada-Iglesias et al., 2011*). The most successful studies, combining this approach with transcriptome data, generated libraries with up to 60% of active enhancers in the target cell type (*Gorkin et al., 2020*; *Graybuck et al., 2021*). Massively parallel reporter assays (MPRAs) have been developed to screen the activity of thousands of sequences simultaneously (*Arnold et al., 2013*; *Inoue and Ahituv, 2015*; *Melnikov et al., 2012*; *White et al., 2013*). However, limitations of sequence synthesis constrain one to choose either a large number of short sequences (e.g., thousands of sequences of 150–250 bp) or a small number of longer sequences (e.g., dozens of sequences of 500–1000 bp) (*Inoue and Ahituv, 2015*). This issue, combined with the difficulty of identifying putative enhancers, leads to a low rate of active enhancers in MPRAs.

Here, we study enhancer location, specificity, and regulatory grammar in melanoma using a variety of MPRA strategies. Melanoma exhibits pronounced heterogeneity within and between patients (*Grzywa et al., 2017*). Two main subtypes or cell states are discernible, the melanocytic (MEL) and the mesenchymal-like (MES) state (*Hoek et al., 2008*; *Hoek et al., 2006*; *Verfaillie et al., 2015*), as well as more recently identified variants of the MEL state, such as the neural crest-like and intermediate states (*Rambow et al., 2018*; *Tsoi et al., 2018*; *Wouters et al., 2020*). The MEL and MES subtypes display distinct epigenomic and transcriptomic profiles, resulting in divergent phenotypes such as migration and therapy response (*Wouters et al., 2020*; *Verfaillie et al., 2015*). Thus, the identification of subtype-specific enhancers may be relevant for therapy, where it could improve safety and efficiency by narrowing down the effect of the treatment to a specific population. Comparisons between MEL and MES yield thousands of regions with differential acetylation (H3K27ac) and accessibility. However, it is unclear which of these subtype-specific regions function as active enhancers and which TFs are responsible for their activity.

In this study, we analyzed regions with MEL- and MES-specific H3K27ac chromatin marks proximal to differentially expressed genes. We designed MPRA experiments to test those regions at three different levels in a panel of patient-derived malignant melanoma (MM) lines. Our results precisely locate the origins of enhancer activity within the larger H3K27ac domains. In addition, we can accurately predict their subtype specificity and ultimately identify a set of rules governing MEL and MES enhancer activity. Furthermore, we show that a melanoma deep learning model (DeepMEL2; *Atak et al., 2021*) trained on ATAC-seq data pinpoints which TFBSs drive enhancer activity and specificity.

## Results
### Design of MPRA libraries based on H3K27ac, ATAC-seq, and synthetic sequences

H3K27ac ChIP-seq peaks are often used for the selection of candidate enhancers (*Creyghton et al., 2010*; *Fox et al., 2020*; *Fu et al., 2018*). However, such peaks can encompass large genomic regions, often 2–3 kb long, while enhancers are usually only a few hundred base pairs in size (*Gasperini et al., 2020*; *Li and Wunderlich, 2017*). To investigate the relationship between H3K27ac signal, chromatin accessibility peaks, and enhancer activity, we designed MPRA libraries at three different levels: 1.2–2.9-kb-sized H3K27ac ChIP-seq peaks, 501-bp-sized ATAC-seq peaks that fall within the H3K27ac regions, and 190-bp subsequences tiling the entire H3K27ac regions.

We designed the H3K27ac ChIP-seq-based library (H3K27ac library) by selecting regions that are specifically acetylated in either the MEL or the MES melanoma cell state and located around differentially expressed genes in a panel of 12 melanoma lines: 3 MES lines (MM029, MM047, and MM099) and 9 MEL lines that cover a spectrum from pure melanocytic to intermediate melanoma (MM001, MM011, MM031, MM034, MM057, MM074, MM087, MM118, SKMEL5) (see Materials and methods, *Figure 1—figure supplement 1a*, *Figure 1a*; *Minnoye et al., 2020*). A special consideration was given to regions overlapping ChIP-seq peaks for SOX10 and MITF (for MEL regions) or AP-1 (JUN and JUNB; for MES regions), known regulators of each state (*Wouters et al., 2020*). A total of 35 MES- and 18 MEL-specific regions, with an average size of 1987 bp, were amplified from genomic DNA (*Supplementary file 1*). Regions were named 'GENE_(-)000X,' where GENE is the associated target gene, (-)000 is the distance to the TSS, and X denotes whether the enhancer is distal, intronic, or at a promoter position (see Materials and methods). The H3K27ac ChIP-seq signal across these selected

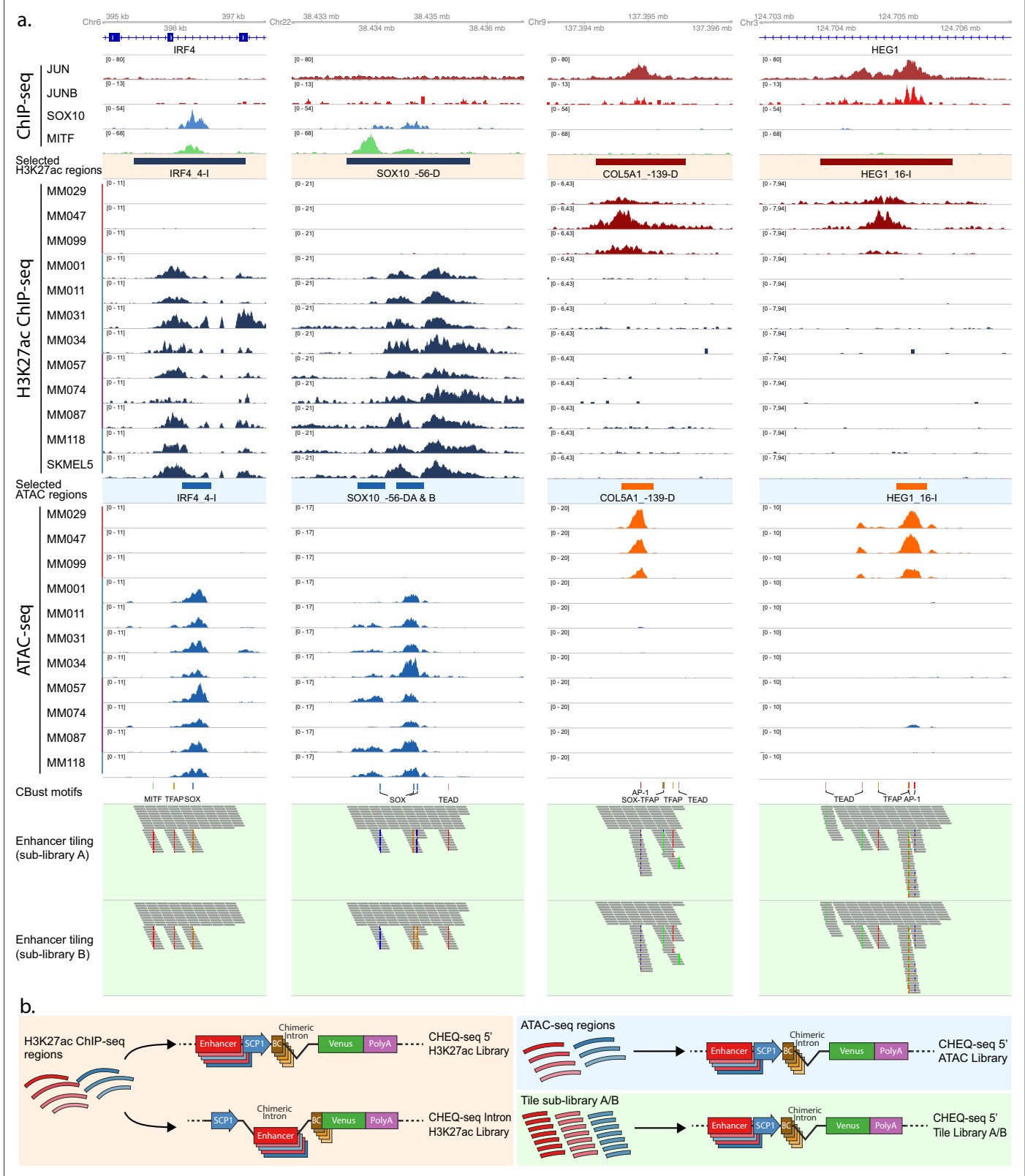

**Figure 1.** Selection and cloning of melanoma state specific regions. (**a**) Cell state-specific regions were selected based on H3K27ac ChIP-seq signal from a panel of melanoma cell lines containing both mesenchymal-like (MES) (red bar) and melanocytic (MEL) lines (pure MEL: blue bar; intermediate MEL: purple bar). ATAC-seq data from the same lines were used to identify accessibility peaks within these regions. Finally, the regions were tiled with 190 bp tiles with a shift of 20 bp (sublibrary A). Sublibrary B was generated by shifting all tiles 10 bp downstream. Cluster-Buster (CBust) was used to

*Figure 1 continued on next page*

*Figure 1 continued*

identify transcription factor (TF) motifs, and new tiles were generated with mutated motifs. (**b**) Reporter vector configurations used for the evaluation of the H3K27ac enhancers (left panel), the ATAC-seq enhancers (top-right panel), and the enhancer tiling (bottom-right panel). SCP1, Super Core Promoter 1; BC, barcode.

The online version of this article includes the following figure supplement(s) for figure 1:

**Figure supplement 1.** Selection pipeline and cell line profiles.

regions displays a good correlation between cell lines of the same subtype and a negative correlation between cell lines of a different subtype (*Figure 1—figure supplement 1b*). This correlation is also observed in the ATAC-seq signal and the target gene expression (*Figure 1—figure supplement 1c, d, and h*). We created two vector libraries based on the CHEQ-seq vector backbone (*Verfaillie et al., 2016*) by cloning the sequences upstream (5′ position) or downstream (intron position) of a minimal promoter (SCP1, see Materials and methods, *Figure 1b*).

Next, we designed a second library consisting of the ATAC-seq peaks contained within the H3K27ac library regions (*Figure 1a*). We used only the H3K27ac peaks that are identified as active in the H3K27ac library or that are assigned as MEL- or MES-specific regulatory regions (respectively represented by regions belonging to topic 4 and topic 7) in our previously published cisTopic analysis of ATAC-seq data from 16 human melanoma cell lines (*Bravo González-Blas et al., 2019*). Each sequence was defined by taking the summit of the ATAC-seq peak and extending 250 bp on each side, resulting in a 501 bp candidate sequence. In total, 28 MES- and 18 MEL-specific ATAC-seq peaks were selected from 37 of the 53 H3K27ac regions (for nine regions, two ATAC-seq peaks were selected). Over the 501 bp of those regions, the correlation of the H3K27ac ChIP-seq, ATAC-seq, and RNA-seq signals between the different cell lines remains the same as for the H3K27ac library (*Figure 1—figure supplement 1e-g*). We cloned the regions in the CHEQ-seq vector, upstream of the SCP1 promoter (*Figure 1b*). In addition, we cloned the same set of sequences in the STARR-seq vector (an alternative MPRA vector) to assess assay-related variability (*Muerdter et al., 2018*).

Finally, to locate the precise origin of enhancer activity, we generated two tiling libraries (A and B, see Materials and methods), encompassing the entire H3K27ac regions. The tiles are 190 bp long with a 20 bp shift between consecutive tiles. The tiling for library A starts at nucleotide position 1 of the H3K27ac ChIP-seq regions while library B starts at position 11, resulting in a final tiling resolution of 10 bp when both sublibraries are taken into account.

We also sought to probe the effect of mutations within putative TFBSs in the selected H3K27ac regions. We used Cluster-Buster (CBust; *Frith et al., 2003*) with position weight matrices (PWMs) for binding sites of key regulators of MEL (SOX10, MITF, TFAP2A) and MES (AP-1 and TEAD) (*Wouters et al., 2020*) to identify candidate TFBSs and generated tiles carrying mutated versions of these motifs (see Materials and methods). For each sublibrary, 800 shuffled tiles were generated as negative controls, resulting in a total of 7412 and 7356 tiles for sublibraries A and B, respectively (*Figure 1a*). Each sublibrary is separately cloned upstream of the SCP1 promoter in the CHEQ-seq vector and is transfected individually (*Figure 1b*, lower-right panel).

## Most MEL-specific acetylated regions harbor enhancer activity in MEL lines

We first transfected all MPRA libraries in the most melanocytic line, MM001 (*Minnoye et al., 2020*; *Wouters et al., 2020*). Of the MEL-specific H3K27ac regions, 75% (14/18) display significant enhancer activity (Benjamini–Hochberg adjusted p-values < 0.05, see Materials and methods) in MM001 with a mean log2 fold change (FC) of 0.23 compared to 26% (8/32) for the MES-specific regions with a mean log2 FC of –1.21 (*Figure 2a*). The activities are consistent across the two library designs (enhancers cloned into the intron or upstream of the TSS; *Figure 2—figure supplement 1a and b*). The ATAC-based library recapitulates these activities, suggesting that the enhancer activity is contained within the sequence of the ATAC-seq peak (*Figure 2b*, *Figure 2—figure supplement 1c–e*). Interestingly, in the five MEL-specific H3K27ac regions where two ATAC-seq peaks were assessed, only one of the two recapitulates the activity of the encompassing region (*Figure 2—figure supplement 2*). This was independently confirmed using the STARR-seq MPRA (*Figure 2—figure supplement 1c*).

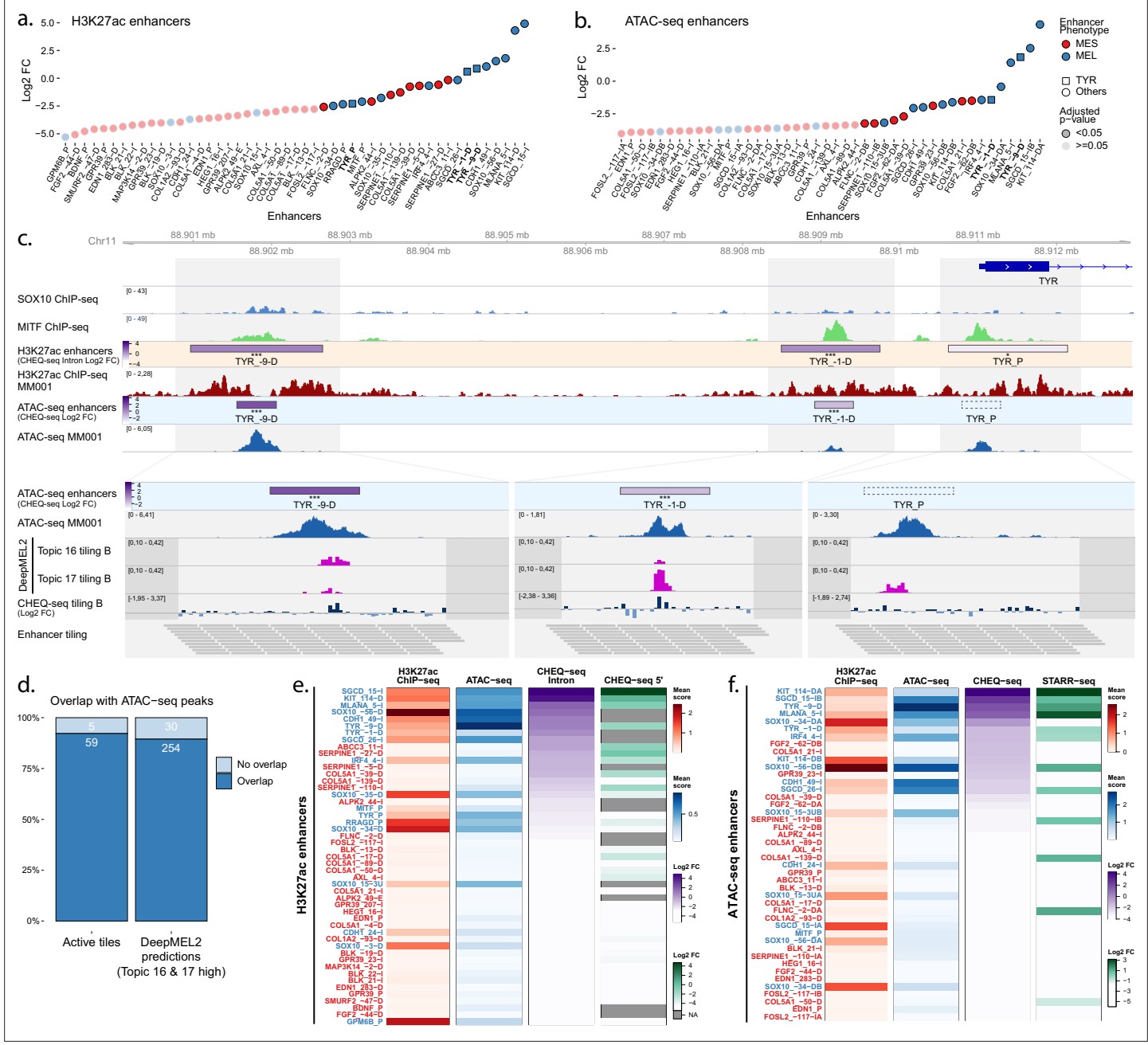

**Figure 2.** Enhancer activity in MM001. (**a, b**) Enhancer activity profile for the CHEQ-seq intron H3K27ac library (**a**) and CHEQ-seq ATAC-seq library (**b**). Enhancer regions displayed in panel (**c**) have their name indicated in bold and their value is displayed with a triangle. (**c**) Enhancer activity of regions selected around the TYR genes. SOX10 and MITF ChIP-seq as well as H3K27ac ChIP-seq and ATAC-seq for MM001 are displayed, and, in the zoomed-in regions (light gray areas), DeepMEL2 predictions and CHEQ-seq values of the enhancer tiling B library are represented. Dark gray areas are regions not covered by a tile. CHEQ-seq activity is visible in the 'H3K27ac enhancers' and 'ATAC-seq enhancers' tracks. Benjamini–Hochberg adjusted p-values: *<0.05; ***<0.001. Dashed box: region not recovered following DNA synthesis, cloning, or massively parallel reporter assay (MPRA). (**d**) Percentage of overlap between active tiles and ATAC-seq peaks (left) and high DeepMEL2 predictions with ATAC-seq peaks (right).(**e, f**) Heatmaps of H3K27ac library (**e**) and ATAC-seq library (**f**), displaying H3K27ac ChIP-seq signal, ATAC-seq signal, and enhancer activity in MM001 ordered by MPRA values. Only MPRA values of significantly active enhancers are displayed.

The online version of this article includes the following source data and figure supplement(s) for figure 2:

**Source data 1.** CHEQ-seq 5' H3K27ac activity.

**Source data 2.** CHEQ-seq intron H3K27ac activity.

**Source data 3.** CHEQ-seq ATAC activity.

*Figure 2 continued on next page*

*Figure 2 continued*

**Source data 4.** CHEQ-seq tiling library B activity.

**Source data 5.** DeepMEL2 prediction scores for tiling library B.

**Source data 6.** STARR-seq ATAC activity.

**Figure supplement 1.** Additionnal MPRAs on H3K27ac and ATAC regions in MM001.

**Figure supplement 2.** Enhancer activity of H3K27ac regions with two selected ATAC peaks in MM001.

Next, we examined the activity of all 190 bp sequences tiled along the entire H3K27ac regions. We confirmed that the majority of active tiles (92.2%) are located within an ATAC-seq peak (*Figure 2c and d*, *Figure 2—figure supplement 2*) and identified active tiles in 7 out of the 10 most active MEL ATAC-seq-based enhancers. Short 190 bp regions can thus often recapitulate the enhancer activity of the larger encompassing region. When two or more consecutive tiles are active, the enhancer may be contained in an even smaller sequence or the activity is coming from independently active elements close to one another.

We recently trained a deep learning model on *cis*-regulatory topics (sets of co-accessible genomic regions clustered by cisTopic; *Bravo González-Blas et al., 2019*) from 30 melanoma lines, called DeepMEL2, that accurately predicts the accessibility and activity of a sequence in the different melanoma subtypes (*Atak et al., 2021*). Each topic used to train the model regrouped accessible regions found in one cell line, in a specific subtype, or in all cell lines. Two topics are associated with the MEL subtype, topics 16 and 17, mostly focused on SOX and MITF motifs, respectively. We scored our 190 bp tiles with DeepMEL2 and found high MEL prediction scores (>0.10, see Materials and methods) specifically within ATAC-seq peaks (*Figure 2d*). Of the top DeepMEL2 predictions for MEL specific topics, 11% (topic 16) and 17.3% (topic 17) are active tiles in the MPRA (with 0.25/0.375 recall and 0.11/0.173 precision for topics 16 and 17, respectively). These low precision values may be explained by the fact that the DeepMEL2 model was trained on ATAC-seq data, thus yielding high prediction scores within ATAC-seq peaks, yet not all of these show positive MPRA activity.

In some cases, we identified multiple acetylated regions near one gene. For the tyrosinase (*TYR*) gene, expressed specifically in MEL lines (*Figure 1—figure supplement 1h*), three regions were selected as MEL-specific and tested at the acetylation, accessibility, and tiling levels (*Figure 2c*). TYR_–9-D and TYR_–1-D regions display high reporter activity, which is subsequently found in the selected ATAC-seq peak. Activity is further narrowed down to tiles corresponding to the DeepMEL2 predictions (*Figure 2c*). The TYR_P region, at the gene's promoter site, has a low activity that is also not found when tiling the region, despite the DeepMEL2 predictions. Those findings suggest that TYR expression in MEL lines is largely dependent on the activity of distal enhancers.

Some other enhancers that are active in the H3K27ac and ATAC-seq libraries are not recapitulated in the tile library (e.g., SOX10_–34-D; *Figure 2—figure supplement 2c*). This can be due to technical reasons, such as the small size of the tiles. Nevertheless, from the combination of the ATAC-seq and enhancer tiling MPRAs, we can conclude that not all subtype-specific ATAC-seq peaks function as a stand-alone enhancer.

We finally compared signals of H3K27ac ChIP-seq mean score, ATAC-seq mean score, and their corresponding MPRA signals (*Figure 2e and f*). With the H3K27ac library (*Figure 2e*), the acetylation signal measured over the selected regions correlates well with the accessibility signal (Pearson's correlation $r$ = 0.77), indicating that ATAC-seq peaks are found within the selected acetylated regions and maintain the same differential signal. Active enhancers are found in the majority (14/18) of the MEL-specific acetylation regions with ATAC-seq signal. This trend is also visible in the ATAC-seq-based library, with most of the active enhancers detected in ATAC-seq peaks (13/19) (*Figure 2f*). However, the moderate correlation between ATAC-seq signal and CHEQ-seq activity in the corresponding library (Spearman's rho = 0.48) also indicates that the peak mean signal is not a good predictor of the activity level of an enhancer. In part, this may be due to confounding of ATAC and H3K27ac read depth by genomic copy number aberrations. Also, the activity displayed by some MES-specific regions lacking ATAC-seq signal in MM001 suggests that closed regions in the genome can still harbor activity in an episomal MPRA.

In conclusion, our enhancer selection resulted in a high rate of active enhancers in MM001 and the design of our MPRA libraries allowed us to precisely pinpoint the origins of, at least part of, the enhancer activity.

## MES-specific H3K27ac/ATAC regions are active in MES lines

Next, we transfected all libraries in the MES line MM029. The activity profiles in this line show that, as expected, the majority of MES enhancers display activity at both the H3K27ac and ATAC-seq level (*Figure 3a and b*, *Figure 3—figure supplement 1a–e*). In regions with two selected ATAC-seq peaks, both MPRA approaches agree on which peak is driving activity (*Figure 3b and d*, *Figure 3—figure supplement 1c*, *Figure 3—figure supplement 2*).

Multiple regions around the Collagen Type V Alpha 1 Chain (*COL5A1*) gene, up to 100 kb upstream of the TSS, were found to be specifically acetylated in MES lines, and we included a total of seven into the library (*Figure 3c.*). Four regions showed significant activity in MM029 (*Figure 3c*). This activity is further confirmed in the ATAC-seq and tiling libraries: the ATAC-seq peaks within the four active H3K27ac regions are all active, while the ATAC-seq peaks within the three negative H3K27ac regions are all negative. In the COL5A1_–17-D region, tiling suggests the presence of three distinct enhancers, including two that are located within small ATAC-seq peaks that were not selected for the ATAC-based library.

DeepMEL2 is trained on both MEL- and MES-accessible regions, and topic 19 has been shown to be the best-performing MES topic (*Atak et al., 2021*). In our data, high topic 19 predictions often overlap with active tiles in MM029 (0.181 precision, 0.537 recall, *Figure 3c and d*). Because several other *cis*-regulatory topics contribute to the prediction of the MES subtype, some tiles do not display a high prediction score for topic 19 despite their activity and are better explained by other topics.

The small shift between each tile and the use of two overlapping libraries provide a high number of measurements throughout the regions, which allows for accurate detection of weakly active enhancers. Such enhancers are found in the SERPINE1_–110-I region, where the SERPINE1_–110-IA ATAC-seq peak is inactive in the CHEQ-seq and STARR-seq assays (*Figure 3d*, *Figure 3—figure supplement 1c*), but the tiling assay shows robust enhancer activity. Of the two ATAC-seq peaks in the FOSL2_–117-I region, neither one recapitulates the activity of the acetylated region. In contrast, the tiling assay reveals clear enhancer activity in both peaks (*Figure 3—figure supplement 2b*).

Similar conclusions to those above for MEL enhancers can now be drawn for MES regions regarding the relationship between H3K27ac, ATAC-seq signal, and enhancer activity (*Figure 3e and f*). ATAC-based and tiling CHEQ-seq assays show that most active enhancers in the H3K27Ac regions reside within ATAC-seq peaks (151/164 active tiles are found in ATAC-seq peaks; 92.1%). Irrespective, ATAC-seq peak mean signal remains a poor predictor of the level of enhancer activity, at least as read out by CHEQ-seq. Moreover, the presence of a differentially accessible ATAC-seq peak does not guarantee enhancer function. Indeed, of 26 differentially accessible peaks, only 14 show demonstrable activity (54%).

In conclusion, MPRAs performed in MM001 and MM029 have shown a high success rate of selected regions to display activity specifically in their corresponding cell state. However, MM001 and MM029 lie at the extremes of the MEL-MES spectrum. To further investigate how the activity of these regions scales along the MEL-MES axis, we studied them in five additional melanoma cell lines, representing more intermediate or transitory melanoma states (*Tsoi et al., 2018*; *Wouters et al., 2020*).

## MES enhancers show lower but consistent activity in intermediate lines

We expanded our panel of cell lines to include two additional MES lines (MM047 and MM099) and three MEL-intermediate lines (MM057, MM074, and MM087). These three lines have high SOX10 and MITF expression (hallmarks of the MEL subtype), yet also show both marker expression and phenotypic characteristics typical for the MES subtype (e.g., AXL expression, TGFb1 signaling activity) (*Tsoi et al., 2018*). Furthermore, in contrast to MM001, when SOX10 expression is lost, these lines shift toward a MES subtype, with expression of AP-1 and increased chromatin accessibility in MES-specific regions (*Wouters et al., 2020*). MM057, MM074, and MM087 were part of the cell lines used for the selection of MEL-specific H3K27ac ChIP-seq regions. As such, they display an acetylation and accessibility profile as well as a transcriptional activity of the associated genes similar to MM001 (*Figure 1—figure supplement 1b–d*). Based on those observations, even though phenotypically different, the

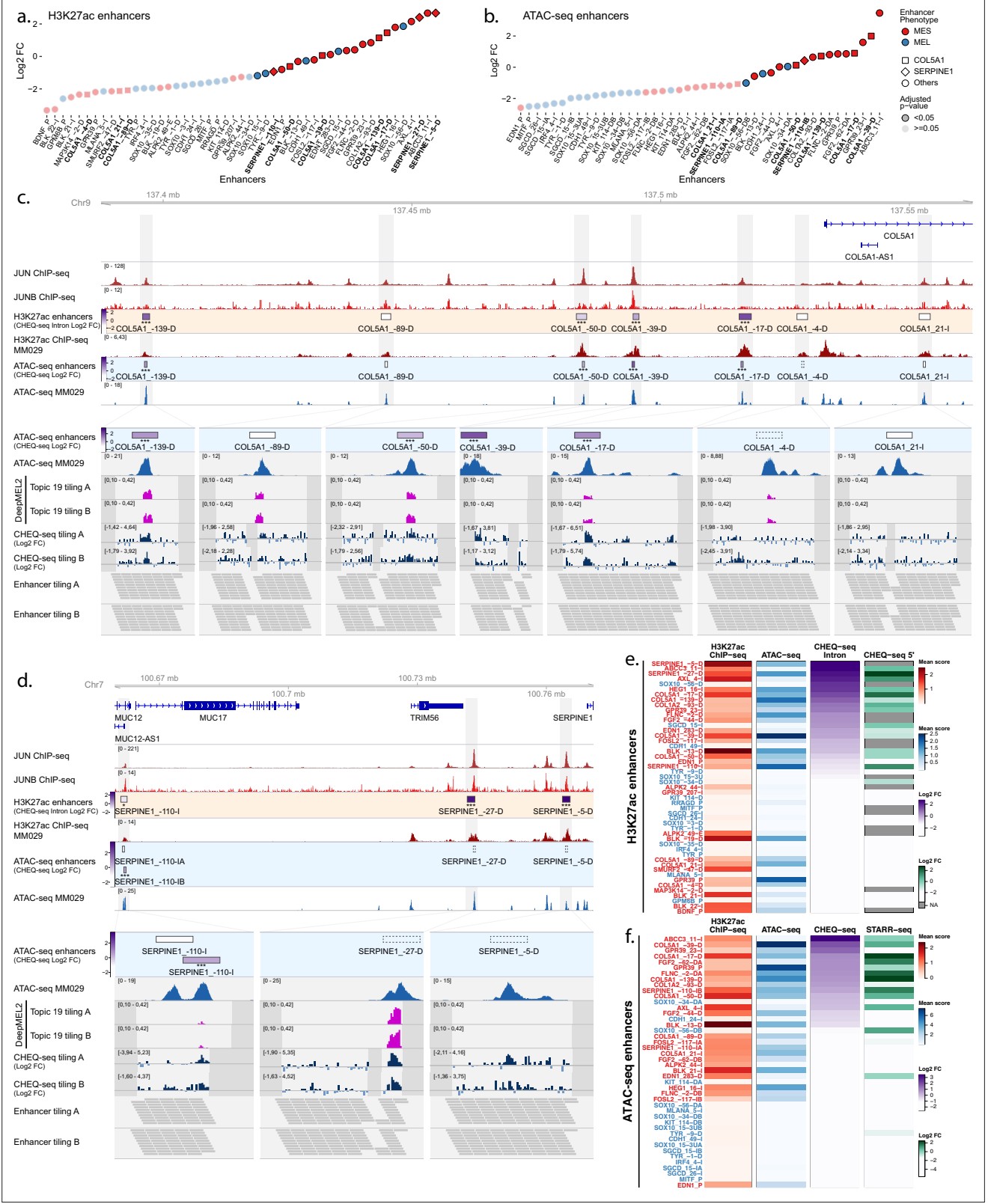

**Figure 3.** Enhancer activity in MM029. Enhancer activity profile for the CHEQ-seq intron H3K27ac library (**a**) and CHEQ-seq ATAC-seq library (**b**). Enhancer regions displayed in panel (**c**) and (**d**) have their name indicated in bold, and their value is displayed with a different shape. Enhancer activity of regions selected around the COL5A1 (**c**) and SERPINE1 (**d**) genes. JUN and JUNB ChIP-seq and H3K27ac ChIP-seq and ATAC-seq for MM029 are displayed, and, in the zoomed-in regions (light gray areas), DeepMEL2 predictions and CHEQ-seq values of the enhancer tiling are represented.

*Figure 3 continued on next page*

*Figure 3 continued*

Dark gray areas are regions not covered by the tiling library. CHEQ-seq activity is visible in the 'H3K27ac enhancers' and 'ATAC-seq enhancers' tracks. Benjamini–Hochberg adjusted p-values: *<0.05; ***<0.001. Dashed boxes: regions not recovered following DNA synthesis, cloning, or massively parallel reporter assay (MPRA). Heatmaps of H3K27ac library (**e**) and ATAC-seq library (**f**), displaying H3K27ac ChIP-seq signal, ATAC-seq signal, and enhancer activity in MM001 ordered by MPRA values. Only MPRA values of significantly active enhancers are displayed.

The online version of this article includes the following source data and figure supplement(s) for figure 3:

**Source data 1.** CHEQ-seq tiling library A activity.

**Source data 2.** DeepMEL2 prediction scores for tiling library A.

**Figure supplement 1.** Additionnal MPRAs on H3K27ac and ATAC regions in MM029.

**Figure supplement 2.** Enhancer activity of H3K27ac regions with two selected ATAC peaks in MM029.

MEL-intermediate lines were expected to have an enhancer profile closely related to what we have observed with MM001.

Indeed, the enhancer activity profile for the H3K27ac library obtained in intermediate lines correlates well with MM001, except for MM074, which moderately correlates with all lines (***Figure 4a***). Interestingly, intermediate lines have the same proportion of active MEL enhancers as MM001 and the same proportion of active MES enhancers as the MES lines (***Figure 4b***). When looking at the mean activity of each enhancer per cell line phenotype (***Figure 4c***), we found a good correlation of MES region activity between all phenotypes (MES vs. MM001, $r = 0.64$; MES vs. intermediate $r = 0.89$). The only difference resides in the strength of the enhancer activity, where MM001 has weak activity and intermediate lines have moderate activity for MES regions.

The comparison of enhancer activity between subtypes highlights the particular case of the SOX10_–56-D region, a MEL-specific H3K27Ac region with enhancer activity in all cell lines (***Figure 4c***). Based on the tiling profile of that region across all tested lines (***Figure 4—figure supplement 1a***), we can identify two active enhancers. One enhancer, located within the largest ATAC-seq peak, is MEL-specific and overlaps with the DeepMEL2 predictions for MEL accessibility (***Figure 4—figure supplement 1a***, green highlight). The other one is located just downstream in a GC-rich region and displays activity in all cell lines (***Figure 4—figure supplement 1a b***, red highlight). This profile explains why the SOX10_–56DB ATAC-seq region, which does not fully cover this GC-rich region, is less active in MES lines (***Figure 2b***; ***Figure 3b***).

With the ATAC-seq region library, the correlation between lines remains the same with a clear separation of the MEL and MES lines and the same pattern in the proportion of active enhancers (***Figure 4d and e***). The overall lower percentage of active MEL enhancers can be explained by the high proportion of regions with two peaks selected, where most of the time only one is active. As in the H3K27Ac library, the MES enhancer activity is higher in intermediate lines than in MM001 but lower than in MES lines (***Figure 4f***).

At the tiling level, intermediate lines show an increased MES enhancer activity comparable to that in the MES lines (***Figure 4g–i***). This results in a reduced correlation with MM001 despite a similarly strong activity of MEL enhancers. As observed in the specific part of the SOX10_–56-D region (***Figure 4—figure supplement 1a***), most active MEL regions have a good specificity, with no activity in MES lines. On the contrary, MES enhancers display a similar activity in MES lines and intermediate lines (***Figure 4i***, ***Figure 4—figure supplement 1c***).

In summary, MEL enhancers display a consistent and specific activity in MEL lines while MES enhancers are not only active in MES lines but also in intermediate lines and to a lesser extent in MM001. This behavior suggests that enhancer activity is driven by the expression of TFs responsible for the MEL and MES subtypes. SOX10, MITF, and TFAP have been identified as drivers of the MEL state and AP-1 and TEAD of the MES state (***Verfaillie et al., 2015***). MM029 expresses MITF at a low level, but this does not result in MEL enhancer activity, suggesting that MITF (or its isoforms in MM029) alone is not sufficient.

## Key transcription factor binding sites explain melanoma enhancer activity

To better understand which TFs are important for MEL enhancer activity, we included tiles with MITF, SOX, and TFAP binding site mutations in our tiling libraries. SOX10 and MITF both represent

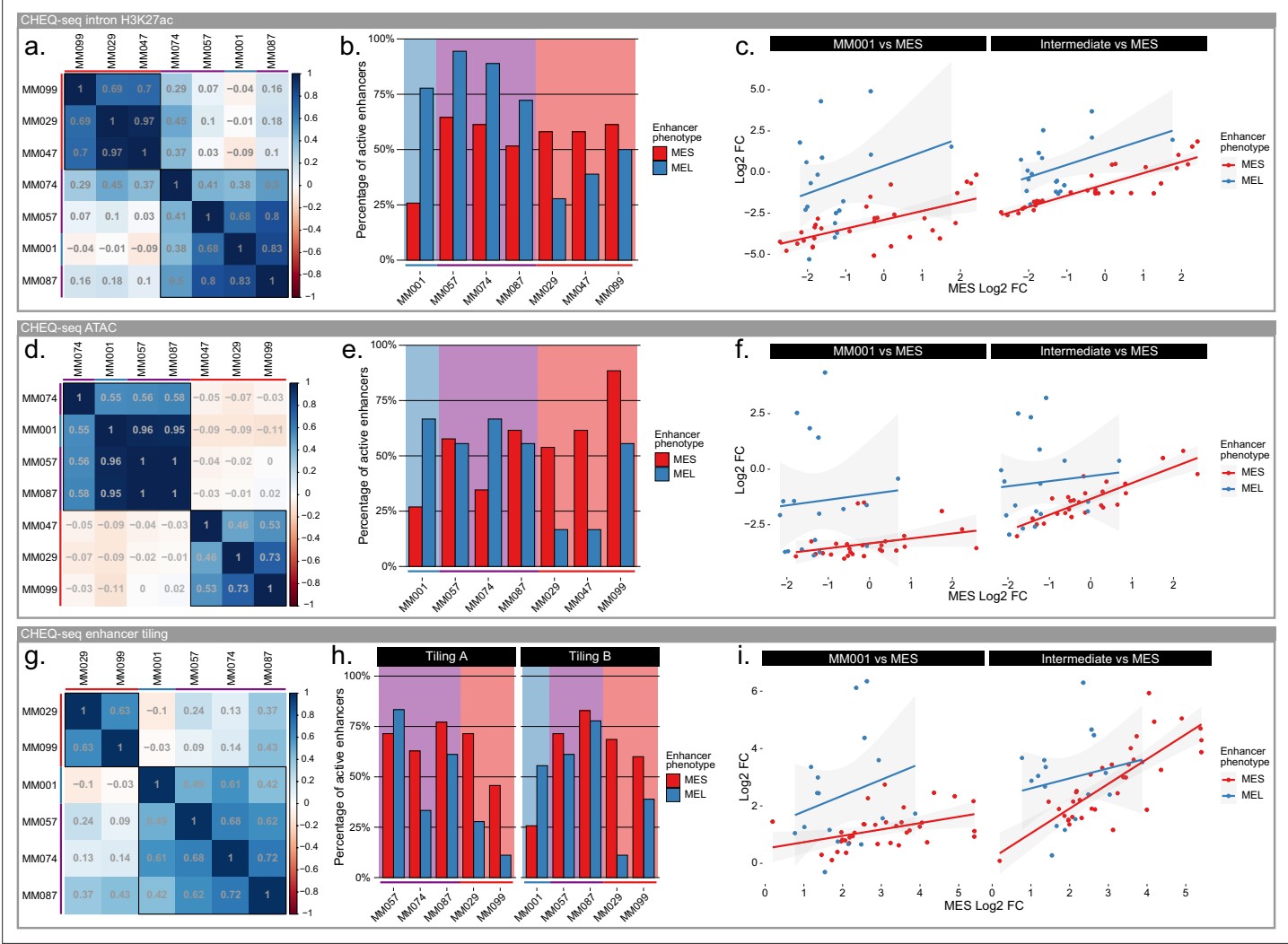

**Figure 4.** Specificity of melanocytic (MEL) and mesenchymal-like (MES) enhancers in intermediate lines. (**a–c**) Pearson's correlation coefficient table across 7 malignant melanoma (MM) lines (**a**), percentage of active MEL and MES enhancers for each line (**b**), and scatterplot of CHEQ-seq results for the CHEQ-seq intron H3K27ac library. (**d–f**) same as panels (**a–c**) but for the CHEQ-seq ATAC-seq library. (**g–i**) same as panels (**a–c**) but for the combined CHEQ-seq enhancer tiling libraries. Red, purple, and blue bars next to cell line names and background indicate MES, intermediate, and MEL lines respectively.

The online version of this article includes the following figure supplement(s) for figure 4:

**Figure supplement 1.** Analysis of SOX10_-56-D and FLNC_-2-D activity in all cell lines.

established lineage TFs in melanocytes and in melanoma, and TFAP2 has been found to co-occur with MITF in active regulatory elements in melanocytes (***Goding, 2000***; ***Harris et al., 2010***; ***Seberg et al., 2017***). SOX10 was previously shown to be necessary for melanoma formation and maintenance by controlling cell survival and cell cycle (***Cronin et al., 2013***; ***Shakhova et al., 2012***). MITF is considered to be a melanoma oncogene (***Garraway and Sellers, 2006***) and has been implicated in various cellular processes (***Goding and Arnheiter, 2019***; ***Levy et al., 2006***).

The comparison of wild-type versus mutant tiles shows that TFAP does not affect the activity of the profiled enhancers in any of the MEL lines (***Figure 5—figure supplement 1***). On the other hand, loss of SOX or MITF significantly affects enhancer activity. KIT_114-D provides a good example of both MITF and SOX motif contributions in MM001 (***Figure 5a***). The tiling library confirms the location of enhancer activity to be in the KIT_114DA region, where both motifs are found to influence the activity. Mutation of the SOX binding site abolishes tile activity. The same is observed when the MITF binding site is mutated, with the notable exception of one tile also containing the SOX motif

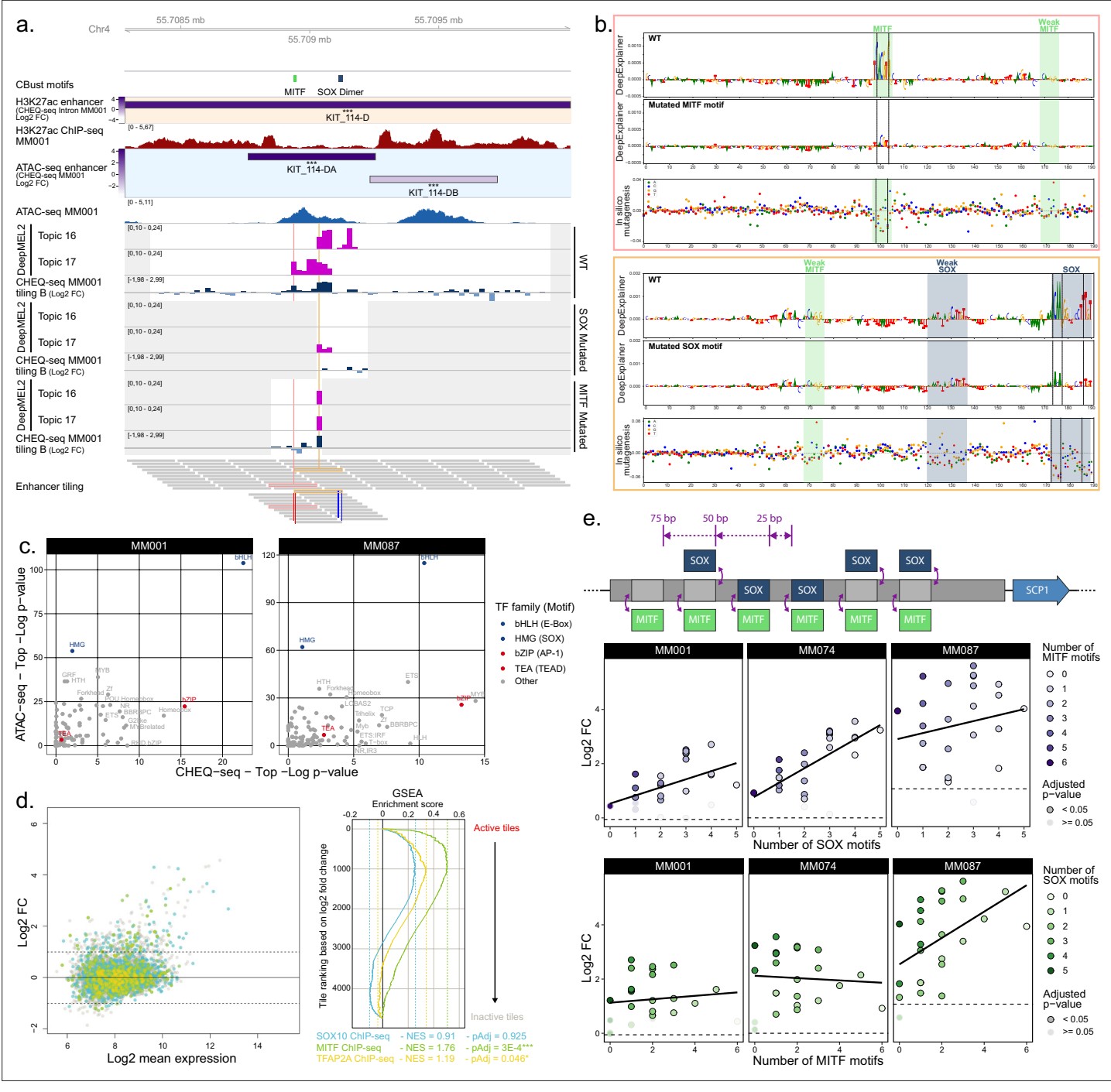

**Figure 5.** Analysis of transcription factors driving MEL specific enhancer activity. (**a**) KIT_114-D region activity summary in MM001. The 'CBust motifs' track shows identified transcription factor binding sites (TFBSs) for MITF and SOX. DeepMEL2 prediction scores for topics 16 (SOX) and 17 (MITF) and CHEQ-seq activity values are shown for the wild-type (WT), SOX mutated, and MITF mutated tiles. Gray areas are regions not covered by the tiling library. The 'Enhancer tiling' track represents the actual location of the tiles. (**b**) DeepExplainer profiles for the tiles highlighted in pink and yellow in the 'Enhancer tiling' track of panel (**a**). Each profile consists of the WT (top panel) and mutated (middle panel) tile nucleotide score and in silico mutagenesis score for the WT tile (bottom panel). Dashed lines indicate the location of mutated nucleotides. (**c**) Comparison of the most enriched motif families as identified by HOMER between ATAC-seq (top ¼ most accessible tiles vs. rest) and CHEQ-seq tiling libraries (all active tiles vs. all inactive tiles) for MM001 and MM087. (**d**) Tiles in the MA plot are colored based on whether or not they overlap with SOX10 (blue), MITF (green), or TFAP2A (yellow) ChIP-seq peaks. Gene Set Enrichment Analysis (GSEA) of the enrichment of SOX10, MITF, and TFAP2A ChIP-seq on the tiles ranked according to their activity (log2 fold change [FC]). For each of the ChIP-seq peak sets, the negative enrichment score (NES) and Benjamini–Hochberg adjusted p-value (pAdj) are shown. (**e**) Top panel: cartoon of SOX and MITF motif combinations in a background sequence. Middle and bottom panels: CHEQ-seq activity

*Figure 5 continued on next page*

*Figure 5 continued*

of synthetic enhancers with background sequence 2 in MM001, MM074, and MM087 sorted by the number of SOX (middle panel) or MITF (bottom panel) motifs present in the sequence. Dashed line indicates the log2 FC value of the background sequence without any motif.

The online version of this article includes the following source data and figure supplement(s) for figure 5:

**Source data 1.** CHEQ-seq SOX-MITF combinations activity.

**Figure supplement 1.** Comparison of wild-type and TFBSs mutant tile activity for key MEL TFs.

**Figure supplement 2.** MITF and SOX10 ChIP-seq signal in differentially acetylated regions.

**Figure supplement 3.** Luciferase assay validation of TF functionality in MM001.

**Figure supplement 3—source data 1.** Luciferase assay values.

**Figure supplement 4.** Investigation of SOX10-dependent ATAC-seq regions in MM087.

**Figure supplement 4—source data 1.** CHEQ-seq SOX10-KD activity.

**Figure supplement 5.** DeepMEL2 prediction scores for SOX-MITF motif combinations.

**Figure supplement 5—source data 1.** DeepMEL2 SOX-MITF combinations prediction scores.

(*Figure 5b*). It is worth noting that despite KIT_114-D(A) being by far the most active region in both the H3K27ac (together with SGCD_15-I) and the ATAC-seq library, tiles with a higher activity have been found in five other MEL regions. This suggests that individual tiles do not necessarily recapitulate the full enhancer activity and specificity coming from the whole accessible regions. The DeepMEL2 topic 16 and 17 prediction scores closely follow tile activity, and both highlight the SOX and MITF motifs (*Figure 5a*). Interestingly, activity and predictions are not centered on the ATAC-seq peak summit. This suggests that, through training with ATAC-seq data, DeepMEL2 has identified TFBSs responsible for both accessibility *and* activity. Motif enrichment analysis on both the active and most accessible tiles in MEL lines identifies the E-box motif (MITF) as highly enriched in both active and accessible sequences while the SOX motif is only enriched in accessible tiles (*Figure 5c*). On the other hand, AP-1 motifs are enriched only in active tiles. The discrepancy between the effect of some SOX mutations on activity and the absence of enrichment for the SOX motif in active tiles could be due to the limited number of MEL enhancers tested (18). Note that MITF and SOX10 motif enrichment agrees with previously published ChIP-seq data, with the most active regions in our MPRA having high ChIP-seq signals (*Figure 5—figure supplement 2a*). Extending our observations to all MEL vs. MES differentially acetylated regions from *Verfaillie et al., 2015*, we found a significant enrichment of both MITF and SOX10 ChIP-seq signals in MEL-specific regions (t-test p-value $< 10^{-16}$, *Figure 5—figure supplement 2b*). Finally, to confirm that the observed effect of mutations on activity is not limited to 190 bp tiles but also applies in the context of 501 bp accessible regions, we mutated MITF or SOX in three active ATAC regions (MLANA_5-I, IRF4_4-I, TYR_–9-D) and tested them in a luciferase assay in MM001 (*Figure 5—figure supplement 3a*; see Materials and methods). For all three regions, mutation of either TFBS from the sequence results in a significant drop in enhancer activity (*Figure 5—figure supplement 3b*). Additionally, mutating ZEB motifs increased activity compared to wild type for both MLANA_5-I and IRF4_4-I.

## SOX10-dependent ATAC-seq peaks with enhancer activity are enriched in MITF motifs

To study MEL enhancers in more detail, we designed a new library. A SOX10 knockdown (KD) with siRNA was performed on MM057 and MM087, shifting these lines to a MES phenotype, and was followed by ATAC-seq after 0, 24, 48, or 72 hr (*Bravo González-Blas et al., 2019*). After cisTopic analysis, we selected 1461 ATAC-seq peaks where accessibility is lost upon KD and tiled them with 190-bp-long sequences and 120 bp shifts, resulting in 6696 individual sequences (*Figure 5—figure supplement 4a–d*). We cloned this library in the CHEQ-seq vector and transfected it in MM087. Analysis of tile activity revealed that 15.1% of the selected ATAC-seq peaks exhibit enhancer function in MM087 (*Figure 5—figure supplement 4c and d*). We performed Gene Set Enrichment Analysis (GSEA) for SOX10 and MITF ChIP-seq in 501mel melanoma line and TFAP2A in human primary melanocytes on the tiles ranked according to their activity (*Figure 5d*). Only the MITF ChIP-seq signal was enriched in active tiles, indicating that SOX10-dependent regions containing MITF sites are preferentially active. As before, these observations were confirmed by differential motif discovery of active

versus inactive tiles with the E-box motif (MITF) among the most strongly enriched motifs (*Figure 5—figure supplement 4e*).

## Synthetic SOX-MITF motif combinations highlight MEL enhancer regulatory grammar

To further investigate a possible interaction between SOX10 and MITF and see if a MEL enhancer can be generated with only those two TFBSs, we designed sequences consisting of combinations of SOX dimer and MITF motifs spread over a 259 bp background sequence (*Figure 5e*, top panel). Twenty-four different combinations were generated in two background sequences, cloned in the CHEQ-seq vector and transfected in MM001, MM074, and MM087. The activity of the enhancer progressively increases with the number of SOX motifs (*Figure 5e*, middle panel). The presence of MITF motifs additionally increases enhancer activity, but this differs from the effect of the SOX motifs (*Figure 5e*, bottom panel). We see a progressive increase of the activity based on the number of MITF binding sites only when there is also a SOX dimer motif present in the sequence. In the absence of a SOX motif, enhancer activity remains low in comparison with sequences containing at least one SOX motif, even when as many as six MITF motifs are present. The presence of multiple SOX motifs greatly increases the enhancer activity, reducing the influence of MITF (*Figure 5e*, bottom panel).

DeepMEL2 predictions on the synthetic sequences confirm these observations (*Figure 5—figure supplement 5a and b*): topic 16 predictions show the same constant increase in score with the number of SOX binding sites. Topic 17 predictions also increase based on the number of MITF motifs when one SOX motif is present and are brought down to background level when SOX is absent. These results highlight a *cis*-regulatory grammar in MEL enhancers involving SOX10 and MITF. As previously described in enhancers regulating pluripotency in mouse embryonic stem cells, regardless of TFBSs positioning, the number of binding sites remains the predominant factor determining activity (*Figure 5—figure supplement 5c*; *King et al., 2020*).

## AP-1 sites can explain MES enhancer activity

Finally, we looked for the presence of AP-1 and TEAD binding sites in the previously tested acetylated regions and generated mutated versions of the enhancer tiles. Comparison of WT versus TEAD or AP-1 mutant tiles shows a positive effect of AP-1 on activity in all cell lines (*Figure 6—figure supplement 1*). On the other hand, TEAD motifs only show a limited effect in the MM029 tiling B sample and MM087 tiling A sample.

The COL5A1_–17-D region illustrates the influence of AP-1 (*Figure 6a*). Within this region, three enhancers are found active in all cell lines bar MM001, which shows activity only for the second enhancer. The enhancer activity of the H3K27ac and ATAC-based regions across the tested MM lines agrees with the tile activity where MM001 has no or low activity and intermediate and MES lines have a strong activity (*Figure 6a and b*). Our previous characterization of the melanoma cell states highlighted AP-1 activity in MEL-intermediate cell lines, which could explain the observed activity of some MES enhancers in MM057, MM074, and MM087 (*Wouters et al., 2020*).

Focusing on the first active region of COL5A1_–17-D, enhancer activity is limited to a subset of the tiles containing an AP-1 motif (*Figure 6c*). Importantly, the observed activity of a tile agrees with its DeepMEL2 prediction score. The generation of DeepExplainer profiles from those predictions reveals the presence of a ZEB repressor motif next to the first AP-1 site (*Figure 6d*; *Postigo and Dean, 1999*). In line with this, enhancer activity is absent from all tiles containing this motif (*Figure 6c and d*). The comparison of WT versus AP-1 mutant tiles specifically in that region further confirms the activator function of AP-1 in MEL-intermediate and MES lines (*Figure 6c–e*). Motif enrichment analysis on both the active and most accessible tiles in MES lines found only AP-1 enriched (*Figure 6f*). AP-1 (JUN and JUNB) ChIP-seq signal is furthermore enriched in H3K27ac regions displaying activity, as well as in MES differentially acetylated regions from *Verfaillie et al., 2015* (t-test p-value < $10^{-16}$, *Figure 6—figure supplement 2*). As for MITF and SOX motifs, we validated the function of AP-1 motifs in three active MES-specific ATAC regions in MM099 via luciferase assay (*Figure 6—figure supplement 3a*). In all three MES ATAC regions, the loss of AP-1 sites results in a reduction of the luciferase activity (*Figure 6—figure supplement 3b*).

The gene regulatory network of the MES subtype was previously found to be mainly controlled by AP-1 and TEAD (*Verfaillie et al., 2015*). Our MPRA results show that only AP-1 has a direct effect

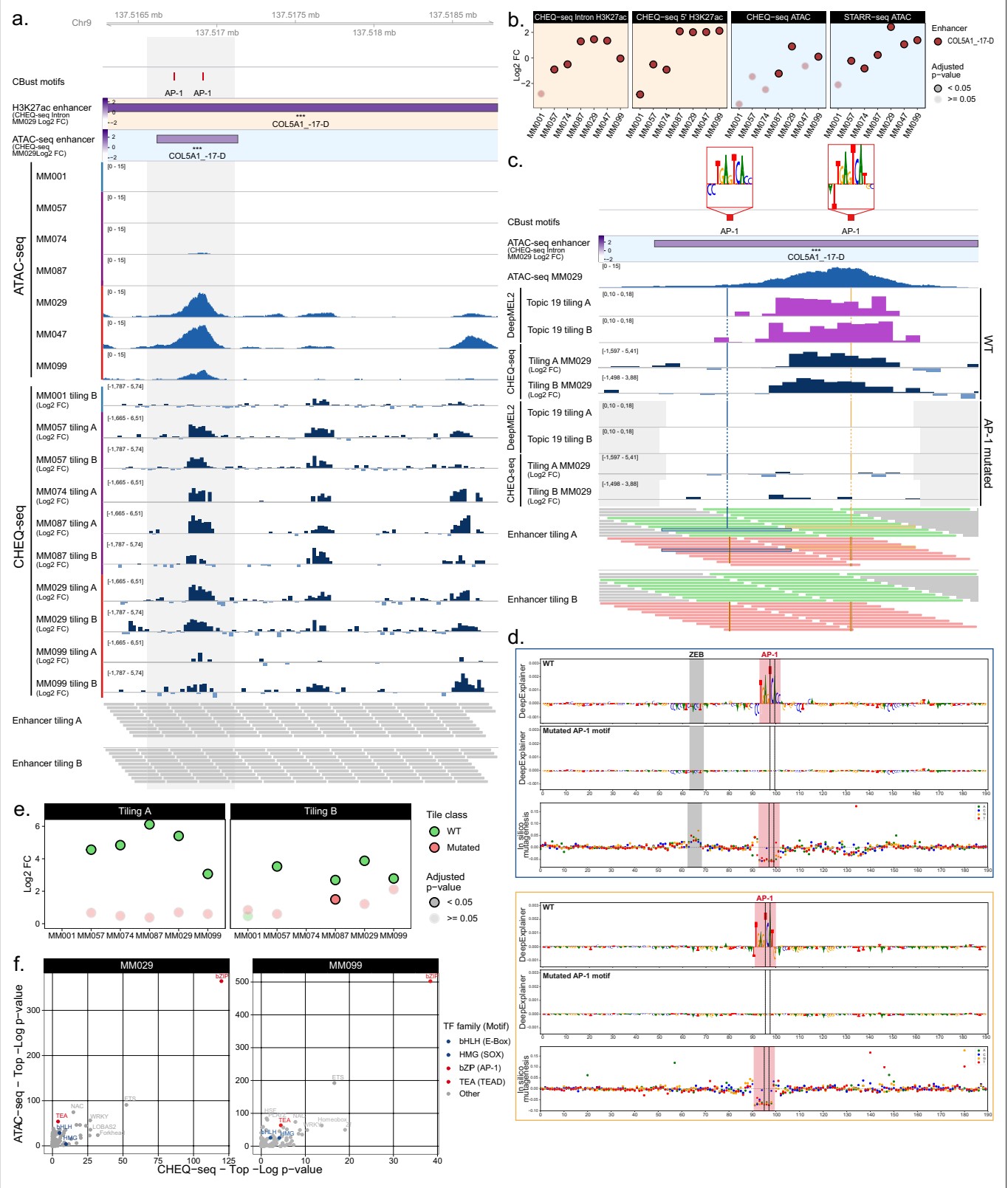

**Figure 6.** Analysis of transcription factors driving MES specific enhancer activity. (**a**) ATAC-seq signal and CHEQ-seq wild-type (WT) tiles activity in the COL5A1_–17-D H3K27ac library region. 'CBust motifs' track shows AP-1 TFBSs. Activity of H3K27ac and ATAC-seq regions is from MM029. (**b**) Enhancer activity for the H3K27ac ChIP-seq and ATAC-based COL5A1_–17-D region. (**c**), CHEQ-seq activity of WT tiles and AP-1 binding site mutant tiles in MM029. 'DeepMEL2' tracks correspond to the accessibility prediction score for each tile. The displayed region corresponds to the highlighted area in

*Figure 6 continued on next page*

*Figure 6 continued*

panel (**a**). Gray areas in 'DeepMEL2' and 'CHEQ-seq tiling' tracks correspond to regions not covered by the tiling library. In the 'Enhancer tiling' tracks, red and green tiles are the mutated and WT tiles, respectively. (**d**), DeepExplainer profiles for the tiles highlighted in blue and yellow in the 'Enhancer tiling A' track of panel (**c**). Each profile consists of the WT (top panel) and mutated (middle panel) tile nucleotide score and in silico mutagenesis score (bottom panel). Dashed lines indicate the location of mutated bases. (**e**), CHEQ-seq signal of the most active tile among the mutated and corresponding WT tiles for the COL5A1_–17-D region. (**f**), Comparison of the most enriched motif families as identified by HOMER between ATAC-seq (top ¼ most accessible tiles vs. rest) and CHEQ-seq tiling libraries (all active tiles vs. all inactive tiles) for MM029 and MM099.

The online version of this article includes the following figure supplement(s) for figure 6:

**Figure supplement 1.** Comparison of wild-type and TFBSs mutant tile activity for key MES TFs.

**Figure supplement 2.** AP-1 ChIP-seq signal in differentially acetylated regions.

**Figure supplement 3.** Luciferase assay validation of TF functionality in MM099.

on enhancer activity in the regions tested, and that the *cis*-regulatory role of TEAD motifs remains unknown.

## Discussion

We have investigated enhancer activity of regions specific to the two main subtypes of melanoma across a panel of patient-derived MM lines. By studying candidate enhancers with different sequence lengths and interpreting the results in conjunction with deep learning predictions, we obtained a better understanding of the regulatory grammar, function, and specificity of melanoma enhancers.

The first studies that used MPRAs were based on very short sequences (84–145 nt) and identified a modular and motif-centric definition of enhancer elements (***Kheradpour et al., 2013***; ***Melnikov et al., 2012***; ***White et al., 2013***). The use of captured genomic regions allowed for longer (hundreds of bp) and more (several millions) sequences to be tested in parallel (***Arnold et al., 2013***; ***Verfaillie et al., 2016***; ***Wang et al., 2018***). While providing a genome-wide view on enhancer activity, the cloned fragments are highly variable in size and do not readily allow the study of corresponding mutant sequences. Furthermore, most studies using MPRAs have been limited to one cell line or two very different cell lines (e.g., K562 and HepG2; or S2 and OSC in *Drosophila*; ***Arnold et al., 2013***; ***Ernst et al., 2016***), leaving the evaluation of enhancer specificity unassessed or reduced to individual examples. Here, we perform MPRAs in a panel of seven cell lines from the same disease, allowing us to validate the cell state specificity of the selected regions.

Our approach of selecting putative enhancers based on specific H3K27ac ChIP-seq peaks, located close to differentially expressed genes, resulted in a high success rate (60–75%), comparable to other studies leveraging a similar approach (***Gorkin et al., 2020***; ***Graybuck et al., 2021***). In contrast, our large-scale library based solely on ATAC-seq peaks that decrease after SOX10 KD contained only 15.1% active enhancers (***Figure 5—figure supplement 4***). This indicates that success rates of enhancer activity vary widely. Whereas most of the differential H3K27ac peaks near marker genes are indeed enhancers, many ATAC-seq peaks scattered across the genome do not act as enhancers when tested in reporter assays, even if they were affected by TF KD. We also find active enhancers in closed chromatin regions, a phenomenon previously described in whole-genome STARR-seq studies in human and *Drosophila* (***Arnold et al., 2013***; ***Liu et al., 2017a***). Those closed regions can often be found accessible in other cell lines of our panel, and their ectopic activity suggests that the chromatin state in the genome differs from the episomal plasmid, keeping the enhancer inactive in the genome despite what seems to be a sufficient TF expression to trigger their activity in our episomal assay. Conversely, some enhancers may rely on 3D chromatin structure and the formation of enhancer-promoter looping to activate transcription via long-range interaction (***Liu et al., 2017b***). This mechanism would require the native chromatin context that cannot be readily recapitulated by our reporter assay.

The influence of the size of the candidate enhancer has previously been assessed, although to a limited extent, and only by choosing sizes that are not representative of chromatin structures (***Klein et al., 2020***). Here, we tested long sequences corresponding to entire H3K27ac regions up to 2.9 kb, 501 bp ATAC-seq peak regions located within the larger H3K27ac domains, and finally, short 190 bp sequences tiling acetylated regions. This revealed that the actual enhancer within a H3K27ac region is usually located within an accessible subregion. Furthermore, the activity of that ATAC-seq peak can often be recapitulated by a sequence of 190 bp in length. Previous studies have used enhancer tiling to

assess the activity of accessible chromatin regions and highlighted the importance of cell type-specific binding sites (*Ernst et al., 2016*; *Wang et al., 2018*). Nevertheless, such a short sequence does not always capture all the regulatory information found within the accessible region. Our detailed description of KIT_114DA and COL5A1_–17-D ATAC-seq peaks highlights the presence of both activator and repressor elements spread over distances larger than 190 bp that cannot be jointly evaluated with our tiled MPRA. This finding highlights the necessity of using sequences ~ 500 bp long to more confidently evaluate enhancers.

Many genes are regulated by an 'array' of enhancers thought to cooperate to regulate target gene expression by forming a chromatin hub (*Di Giammartino et al., 2020*; *Gorkin et al., 2014*; *Zhu et al., 2021*). For several loci in this study, we selected multiple regions near the same target gene. This experiment indicated that not all regions with a similar acetylation and ATAC-seq profile harbor enhancer activity in the reporter assay. As previously described by genomic editing of enhancers displaying similar chromatin features, enhancer cooperativity can be required to activate transcription (*Hnisz et al., 2015*; *Huang et al., 2016*). This makes the identification of relevant TFBS the most informative method to find active enhancers. In that regard, the use of deep learning models like DeepMEL2 (*Atak et al., 2021*), even when they are trained only on ATAC-seq data, provides a powerful tool to identify key TFs and evaluate enhancer activity.

Intriguingly, we find a gradient of MES enhancer activity across our panel of cell lines, with MM001, intermediate, and MES lines having low, medium, and high activity, respectively. These enhancer profiles do not agree with their acetylation, ATAC-seq, and transcriptome profiles, which all show a strong intra-subtype homogeneity. This discrepancy was more pronounced with short 190 bp sequences.

Our results suggest that AP-1 is responsible for the activity of MES enhancers, agreeing with its predominance in allele-specific chromatin accessibility variants in melanoma (*Atak et al., 2021*). Note that AP-1 is also typically activated during the stress response (*Hess et al., 2004*). This may contribute to the limited activity of MES regions in MM001 following electroporation. Nevertheless, AP-1 enhancers show much higher activity in intermediate lines, which cannot be explained by a stress response alone. The observed activity of AP-1 enhancers in intermediate lines, even though they are neither accessible nor acetylated in these lines, could be due to the episomal nature of our reporter assay, whereby a (limited) AP-1 activity, as observed previously (*Wouters et al., 2020*), may be sufficient to activate enhancers on a plasmid, but not in the endogenous locus. The use of a genome integrated reporter assay could help determine if the accessibility of the sequence is required for MES enhancer activity in MEL lines.

The analysis of MEL regions revealed that both SOX and MITF binding sites independently contribute to enhancer activity. TFAP binding sites did not show any influence on enhancer activity in the tested H3K27ac regions and showed only a small enrichment in SOX10-dependent ATAC-seq peaks displaying activity (NES: 1.19; pAdj: 0.046). These results contradict previous observations of TFAP2 and MITF co-occurrence in active regulatory elements in melanocytes and of TFAP2 pioneer function during cell fate commitment in the neural crest (*Rothstein and Simoes-Costa, 2020*; *Seberg et al., 2017*). These differences may be explained at least in part by the limited number of regions selected in our H3K27ac library or by a possible reduced influence of TFAP2 in the melanoma lines used in this study. Interestingly, we identify an interaction between SOX and MITF where the presence of at least one SOX dimer motif greatly increases the activity of enhancers containing MITF motifs. It is not clear if this interaction is due to the direct cooperation of SOX10 and MITF or the pioneer factor function of SOX family members (*Julian et al., 2017*). Cooperativity between these two TFs could be governed by strict rules requiring the presence of a SOX motif for activity or could be a more 'loose' requirement for heterotypic clusters of TFBSs (i.e. containing binding sites for multiple TFs) to maintain a high level of activity. Indeed, studies have shown that, in some cases, homotypic clusters of TFBSs yield lower expression than heterotypic clusters of TFBSs (*Fiore and Cohen, 2016*; *Levo and Segal, 2014*; *White et al., 2016*). While our exploration of synthetic SOX/MITF-based enhancers did not include sequences with fewer than six MITF and no SOX motifs, our enhancer tiling MPRA and luciferase assay did reveal such sequences to be active (e.g., KIT_114-D and SOX10_–35-D; *Figure 2—figure supplement 2b and c*; MLANA_5-I; *Figure 5—figure supplement 3b*). This suggests that a SOX motif is not required in enhancers with few MITF motifs. Alternatively, the pioneer function of SOX would suggest that the sequences cloned in the reporter vector are at least partially

chromatinized. Riu et al. previously presented evidence supporting plasmid chromatinization, more precisely in regions of bacterial origin, resulting in silencing of a human expression cassette (*Riu et al., 2007*). However, it is still unclear if non-coding human regions could also be subject to chromatinization in a plasmid context and what role sequence length might play. We find a less pronounced influence of the SOX motif in short 190 bp compared to 259 bp sequences, and shorter MES enhancers also show more ectopic activation in intermediate lines, which may suggest that short enhancer sequences are being less efficiently chromatinized. Consequently, short enhancers might be characterized by increased accessibility in an extrachromosomal reporter, uncoupling measured activities from those demonstrated at the endogenous locus. Kheradpour et al. proposed a similar hypothesis by suggesting that DNA sequence features contained within the tested elements are partly responsible for establishing the endogenous chromatin state (*Kheradpour et al., 2013*). These considerations are also relevant in the context of extrachromosomal DNA driving oncogene overexpression in various human malignancies (*Verhaak et al., 2019*). The use of integrated MPRA or the measurement of chromatin accessibility in the plasmid would help to determine if enhancer sequences are subject to chromatin modification.

With this study, we show that melanoma subtype-specific enhancers can be identified, even down to sizes of 190 bp. This opens up promising avenues to use information gathered from MPRAs to identify small, specific enhancers for enhancer and gene therapies. Our assay combining SOX and MITF motifs showed that new MEL enhancers can be designed by addition of the corresponding TFBSs to a random background sequence and that the level of activity can be precisely controlled by varying the number of TFBSs. Using genomic regions confirmed as MEL enhancers and adjusting the motif sequences and numbers could further improve enhancer design.

## Materials and methods

### Key resources table

| Reagent type (species) or resource | Designation | Source or reference | Identifiers | Additional information |
|---|---|---|---|---|
| Commercial assay or kit | KAPA-HiFi HotStart ReadyMix | Roche | Material # 7958935001 | |
| Commercial assay or kit | NEBuilder HiFi DNA Assembly Master Mix | NEB | Cat # E2621L | |
| Commercial assay or kit | In-Fusion HD Cloning Plus Kits | Takara | Cat # 638,911 | |
| Recombinant DNA reagent | pSA293-CHEQ-seq | This paper | Addgene: 174669; RRID:Addgene_174669 | CHEQ-seq backbone vector |
| Other | MF-Millipore 0.05 µm | Merck | Ref: VMWP02500 | 25 mm diameter, mixed cellulose esters (MCE) membrane |
| Strain, strain background (*Escherichia coli*) | Endura Electro Competent Cells | Lucigen | Cat # 60242-2 | Electrocompetent bacteria |
| Commercial assay or kit | MinElute PCR Purification Kit | QIAGEN | ID: 28004 | |
| Commercial assay or kit | SPRIselect | Beckman Coulter | Product # B23317 | |
| Commercial assay or kit | innuPREP RNA Mini Kit 2.0 | Analytik Jena | Cat # 845-KS-2040250 | |
| Commercial assay or kit | Dynabeads mRNA purification kit | Ambion | Cat # 61006 | |
| Commercial assay or kit | GoScript RT kit | Promega | Cat # A5003 | |

*Continued on next page*

*Continued*

| Reagent type (species) or resource | Designation | Source or reference | Identifiers | Additional information |
|---|---|---|---|---|
| Recombinant DNA reagent | pGL4.23-GW | Promega | Addgene: 60323; RRID:Addgene_60323 | Luciferase backbone vector |
| Recombinant DNA reagent | pRL-TK | Promega | Cat # E2241 | Renilla vector |
| Commercial assay or kit | Dual-Luciferase Reporter Assay System | Promega | Cat # E1910 | |

## MPRA design and cloning

### H3K27ac-based library

The selection of H3K27ac ChIP-seq regions harboring potential state-specific enhancer activity was done as follows (*Figure 1—figure supplement 1a*). Differentially enriched H3K27ac ChIP-seq peaks between MEL (6669) and MES (13,453) state, identified in our previous study (*Verfaillie et al., 2015*), were used as input in i-cisTarget (*Imrichová et al., 2015*) to identify enriched TFBSs in specifically acetylated regions. For each melanoma subtype, the target genes of the top three enriched TFBSs were extracted and used to filter the list of differentially expressed genes identified in each subtype (*Verfaillie et al., 2015*). Those genes were ordered by differential expression in MES vs. MEL lines, and the TSS of the top and bottom genes was used as the starting point to manually search for H3K27ac regions specific for MES and MEL, respectively. ChIP-seq tracks are visually investigated using the UCSC Genome Browser for regions displaying acetylation peaks only in one state close to those genes. Identified regions, within ~150 kb of the TSS, are associated with the differentially expressed gene, even if some of them are closer to or within another gene. Candidate regions were preferentially selected if they also overlapped with ChIP-seq peaks for SOX10 or MITF for MEL regions and JUN and JUNB (AP-1) for MES regions. 65 regions with a size <3 kb were manually selected, and primers to amplify them were designed using Primer3Plus within the flanking first 100 bp on each side of the sequence. 15 bp extensions were added to the primers to allow recombination with the region upstream of the SCP1 promoter in the CHEQ-seq vector (*Verfaillie et al., 2016*) via In-Fusion reaction (primer list is given in *Supplementary file 2*). Regions were named as follows: GENE_(-)000X with 'GENE' the name of the gene associated to the region, (-)000 the distance (in kb) between the gene TSS and the highest peak summit (H3K27ac) of the region, and 'X' the gene annotation of the region with 'P' for promoter, 'I' for intron, 'E' for exon, 'D' for distal, and '3U' for 3' UTR.

All PCR amplifications performed in this study make use of the KAPA HiFi HotStart ReadyMix (Roche, Basel, Switzerland). The amplification of the selected regions was done from MM074 genomic DNA in a 50 µl reaction. PCR fragments were then purified on a 0.8% agarose gel and the sequence was confirmed via Sanger sequencing. 53 regions could be successfully amplified (*Supplementary file 1*). For the generation of the CHEQ-seq 5' library, the CHEQ-seq vector containing a random 17 bp barcode (BC) upstream of the synthetic intron was linearized by inverse PCR with the primers 'CHEQ-seq_lin_5'_For' and 'CHEQ-seq_lin_5'_Rev,' resulting in a fragment with both ends overlapping with the primers designed to amplify the selected regions. Amplified regions were mixed in equimolar ratio and introduced in the CHEQ-seq vector (Addgene #174669) via In-Fusion reaction (Takara Bio, Kusatsu, Japan) with a vector to insert ratio of 1:2.

For the generation of the CHEQ-seq intron library, amplified regions were reamplified with primers containing adaptors to allow recombination within the intron of the CHEQ-seq vector ('H3K27Ac_lib_intron_For', 'H3K27Ac_lib_intron_Rev'). The CHEQ-seq vector containing a random 18 bp BC downstream of the synthetic intron was linearized by inverse PCR with the primers 'CHEQ-seq_lin_Intron_For' and 'CHEQ-seq_lin_Intron_Rev.' Reamplified regions were mixed in equimolar ratio and introduced in the CHEQ-seq vector via In-Fusion reaction with a vector to insert ratio of 1:2. The In-Fusion reactions were dialyzed against water in a 6 cm Petri dish with a membrane filter MF-Millipore 0.05 µm (Merck, Kenilworth, NJ) for 1 hr. Reactions were recovered from the membrane, and 2.5 µl of the reaction was transformed into 25 µl of Lucigen Endura ElectroCompetent Cells (Biosearch

Technologies, Hoddesdon, UK). Transformed bacteria were cultured overnight in a shaker before maxiprep.

## ATAC-based library

The sequences constituting the ATAC-based library were selected from ATAC-seq peaks from MM001, MM029, MM047, MM057, MM074, MM087, and MM099 overlapping with H3K27ac library regions. Among this subset, only peaks that overlap with regions identified as active in the H3K27ac library or that are assigned as MEL- or MES-specific regulatory regions (respectively represented by regions belonging to topics 4 and 7) in our previously published cisTopic analysis of ATAC-seq data from 16 human melanoma cell lines were retained (*Bravo González-Blas et al., 2019*). The final selection included 46 ATAC-seq peaks (*Supplementary file 1*). When two ATAC peaks were selected within the same H3K27ac region, an 'A' and a 'B' were added to the region name to distinguish them (e.g., KIT_114DA and KIT_114DB). The summit of each peak was extended by 250 bp on both sides to generate a 501 bp sequence. At the 3′ end of the sequence, a 16 bp spacer sequence was added, followed by an 8 bp BC specific for each selected region. 38 sequences were synthesized by Twist Bioscience (South San Francisco, CA). This includes four sequences where an A or a T was substituted to break a polyA or polyT, respectively, in order to make the sequences compatible with the synthesis process. Primers were designed for the remaining eight sequences to amplify them from genomic DNA ('ATAC-seq-based library' in *Supplementary file 2*). Only one PCR, that is, for the amplification of the SERPINE1_–27-D sequence, was not successful. The CHEQ-seq vector containing a random 17 bp BC upstream of the synthetic intron and the STARR-seq ORI vector (Addgene #99296; *Muerdter et al., 2018*) were linearized via inverse PCR ('ATAC-seq-based library' in *Supplementary file 2*). The individual sequences were pooled together in equimolar ratio, and NEBuilder (New England Biolabs, Ipswich, MA), with a vector to insert ratio of 1:2, was used to introduce them in the vectors. Dialysis and transformation were performed similarly to the H3K27ac library. Before culture for maxiprep, 1:100,000 of the transformed bacteria was plated on an LB-agar dish with carbenicillin to estimate the complexity of the cloned library. A volume of bacteria corresponding to a complexity of 3500 BCs per enhancer was put in culture for maxiprep.

## Enhancer tiling libraries

Using a custom script, tiles were generated by selecting 190 bp from the start (library A) or position 11 (library B) of the H3K27ac selected regions and by switching every 20 bp in 3′ direction. Tile generation is stopped when the position of the final nucleotide of the tile is superior to the final nucleotide of the H3K27ac region. To generate mutated tiles, we first scanned all selected H3K27ac regions with CBust (*Frith et al., 2003*) and the following PWM separately: transfac_pro__M08838 (SOX10 dimer); homer__RTCATGTGAC_MITF; transfac_pro__M01859 (TFAP monomer); tfdimers__MD00038 (TFAP dimer); tfdimers__MD00591 (SOX10-TFAP dimer); hocomoco__JUN_f1; homer__NATGAST-CABNN_Fosl2; cisbp__M5907 (TEAD). Using a custom script, motifs were mutated on their two or four most important nucleotides for monomers and dimers, respectively. In order to keep the number of tiles manageable, only highly conserved motifs were mutated. This resulted in some potentially functional sites (presence of TF ChIP-seq peak) not being recognized by PWM scores. Mutated tiles for each motif were generated separately so that all occurrences of a single motif are present. Shuffled negative control tiles were generated by shuffling all WT and mutated sequences with uShuffle (*Jiang et al., 2008*). Sequences containing a stretch of the same nucleotide for six or more nucleotides were filtered out. The remaining tiles were scored with CBust using the same PWMs and parameters as before, and 800 tiles containing no motifs were selected at random. In total, libraries A and B contained 7412 and 7393 tiles, respectively (*Supplementary file 1*).

Adaptor sequences were added to the tiles: 'Adaptor_LibA_5′' and 'Adaptor_LibA_3′' for library A and 'Adaptor_LibB_5′' and 'Adaptor_LibB_3′' for library B. The use of different adaptors for each library will result in the insertion of the library B 20 bp downstream of the library A, providing a slightly different surrounding context that combined with a high number of barcodes per enhancer aims at reducing experimental noise. Final libraries were synthesized via Agilent's Oligonucleotide Library Synthesis Technology (Santa Clara, CA).

Oligonucleotide libraries were resuspended in endotoxin-free TE buffer pH 8 to a final concentration of 20 nM. For each library, 10 PCR reactions were performed with 2 µl of resuspended library

for 12 cycles with primers 'Lib_A_amp_For' and 'Lib_A_amp_Rev' or 'Lib_B_amp_For' and 'Lib_B_amp_Rev.' The PCR product was first cleaned up using MinElute (QIAGEN) with five PCR reactions per column, then pooled together and cleaned up a second time with 1.6× SPRI beads (Beckman Coulter, Brea, CA). The CHEQ-seq vector containing a random 17 bp BC upstream of the synthetic intron was linearized via inverse PCR with primers 'CHEQseq_lin_A_For' and 'CHEQseq_lin_A_Rev' or 'CHEQseq_lin_B_For' and 'CHEQseq_lin_B_Rev.' Amplified libraries and the corresponding linearized vector were combined in an NEBuilder reaction with a vector to insert ratio of 1:3.25. Dialysis, transformation, and maxiprep were performed similarly to the H3K27ac library.

## SOX10 knockdown-based library

To define a set of SOX10-dependent MEL enhancers, we used public OmniATAC-seq data during a time series of SOX10 knockdown (0, 24, 48, 72 hr) on two melanoma cells lines (MM057 and MM087) (GSE114557; *Bravo González-Blas et al., 2019*). We generated 50 simulated single cells per condition by randomly sampling 50,000 reads per cell. Candidate regulator regions were defined by peak calling with MACS2 (v.2.0.10) in each of the bulk samples and merging the condition-specific peaks using mergeBed (part of BEDtools, v.2.23.0), and blacklisted regions were removed using https://sites. google.com/site/anshulkundaje/projects/blacklists (hg19). We ran cisTopic (v0.2.1; *Bravo González-Blas et al., 2019*) (parameters: $\alpha$ = 50/T, $\beta$ = 0.1, burn-in iterations = 500, recording iterations = 1000) for models with a number of topics between 2 and 25. The best model was selected on the basis of the highest log-likelihood, resulting in 16 topics. We binarized the topics using a probability threshold of 0.99 and continued with topic 11, which contained regions that are accessible in baseline conditions but lose accessibility following SOX10 knockdown. A total of 1461 enhancers were chosen to be tested. To increase the resolution of the library, we tiled the sequences to 190 bp using a 120 bp sliding window across the enhancers, resulting in 6696 tiles. In addition, 100 shuffled negative control sequences were generated similarly to the tiling libraries (*Supplementary file 1*). Vector-specific adapters were added to the sequences, and the tiles were synthesized via Agilent's Oligonucleotide Library Synthesis Technology.

The library was amplified via PCR using primers 'Lib_A_amp_For' and 'Lib_A_amp_Rev' and cloned into the linearized CHEQ-seq plasmid by following the same procedure as for the enhancer tiling library A. Dialysis, transformation, and maxiprep were performed similarly to the H3K27ac library.

## Synthetic combinations of SOX and MITF library

Random 259 bp sequences were generated using SMS2-Random DNA Sequence tool (*Stothard, 2000*). Two sequences, displaying no enrichment in any of the topics defined in our previously published cisTopic analysis (*Bravo González-Blas et al., 2019*), were selected as background sequences. SOX and MITF motifs (ACAAAGACGGCTTTGT and CACGTG, respectively) were inserted in the sequence with a motif in the center and the other motifs placed upstream and downstream with a distance of 25, 50, or 75 bp (*Figure 5e*, top panel). A complete list of motif combinations can be found in *Supplementary file 1*. 200 negative control shuffled sequences were generated with uShuffle as described previously. A 11 bp BC specific for each enhancer was placed in 5' position of the sequence. Barcoded enhancers were finally flanked with the adaptors GAGCATGCACCGGTG and CGCTTCGAGCAGACA in 5' and 3', respectively. The final library was synthesized by Twist Bioscience as an Oligo Pool. The oligonucleotide library was resuspended according to the manufacturer's recommendation and amplified via PCR with the primers 'CHEQ_comb_amp_For' and 'CHEQ_comb_amp_Rev.' The CHEQ-seq vector with a 17 bp random BC upstream of the intron was linearized via inverted PCR with the primers 'CHEQ_comb_lin_For' and 'CHEQ_comb_lin_Rev.' NEBuilder was then used to combine the library with the linearized vector with a vector to insert ratio of 1:2. Dialysis and transformation were performed similarly to the H3K27ac library. Before culture for maxiprep, 1:100,000 of the transformed bacteria was plated on an LB-agar dish with carbenicillin to estimate the complexity of the cloned library. A volume of bacteria corresponding to a complexity of 500 BCs per enhancer was put in culture for maxiprep.

## Enhancer–barcode assignment

### CHEQ-seq 5′/intron for H3K27ac ChIP-seq regions

The part of the plasmid extending from the enhancer till the random BC was amplified via PCR with primers 'PacBio_5′_For,' 'PacBio_5′_Rev,' 'PacBio_Intron_For,' and 'PacBio_Intron_Rev' for CHEQ-seq 5′ and CHEQ-seq intron, respectively. Gel extraction was performed to isolate the PCR product with the correct size range using the NucleoSpin Gel and PCR Clean-up kit (Macherey-Nagel, Düren, Germany). PacBio sequencing library preparation for both libraries was done by the Genomics Core Leuven (KU Leuven). Sequencing was done with a PacBio Sequel for long-read sequencing (Pacific Biosciences, Menlo Park, CA) with both libraries sequenced with one SMRT cell. We obtained 66,407 and 105,590 reads with three passes for CHEQ-seq 5′ and CHEQ-seq intron, respectively.

Enhancer–BC assignment was done with a custom script. Briefly, enhancers and random BCs were independently extracted from the reads with Cutadapt (*Martin, 2011*). Enhancer sequences were mapped with Minimap2 (*Li, 2018*) and a custom genome containing all the cloned regions, and only MAPQ ≥ 4 were retained. Mapped enhancers were linked back to random BCs of the vector. Following assignment, 46 (86.8%) and 50 (94.3%) sequences could be identified in the CHEQ-seq 5′ and intron library, respectively. The CHEQ-seq 5′ library displayed an average of 31.9 BCs per enhancer while the intron library displayed an average of 604.5 BCs per enhancer.

### CHEQ-seq for ATAC-seq regions

A PCR amplification of the enhancer-specific BC, together with the random BC, was done with the primers 'Enh-BC_ATAC_Stag = X_For' and 'Enh-BC_ATAC_Rev.' Illumina sequencing adaptors were added during a second round of PCR with the primers 'i5_Indexing_For' and 'i7_Indexing_Rev.' After sequencing in NovaSeq600 for 50 cycles in read 1 and 49 cycles in read 2, enhancer BCs and random BCs were extracted from read 1 and read 2, respectively, with Cutadapt before being filtered for quality ($Q > 30$). Following assignment, 44 (95.7%) sequences could be identified, with an average of 34,815 BCs per enhancer.

### CHEQ-seq for enhancer tiling/SOX10-KD library

Cloned sublibraries were amplified via PCR with the primers 'Enh-BC_Tiling-A_Stag = X_For' (for sublibrary A and SOX10-KD library), 'Enh-BC_Tiling-B_Stag = X_For' (for sublibrary B), and 'Enh-BC_Tiling_Rev.' Illumina sequencing adaptors were added during a second round of PCR with the primers 'i5_Indexing_For' and 'i7_Indexing_Rev.' After sequencing in NovaSeq600 for 251 cycles in read 1 and 51 cycles in read 2, whole-length enhancers and random BCs were extracted from read 1 and read 2, respectively, with Cutadapt before being filtered for quality ($Q > 30$). Enhancer reads were mapped and linked to random BCs as previously described for CHEQ-seq 5′/intron for H3K27ac ChIP-seq regions. Following assignment, 7356 (99.2%), 7344 (99.8%), and 6773 (99.7%) sequences could be identified in the sublibraries A and B and the SOX10-KD library, respectively, with an average of 3096, 3021, and 8056 BCs per enhancer.

### CHEQ-seq for SOX-MITF synthetic combinations

This CHEQ-seq library was prepared for sequencing similarly to the CHEQ-seq enhancer tiling A sublibrary. After sequencing in NovaSeq600 for 50 cycles in read 1 and 49 cycles in read 2, enhancer BCs and random BCs were extracted from read 1 and read 2, respectively, with Cutadapt before being filtered for quality ($Q > 30$). Following assignment, 249 (99.6%) sequences could be identified, with an average of 276 BCs per enhancer.

## Cell lines

The cell lines used in this study were patient-derived melanoma lines obtained from the laboratory of Pr. Ghanem-Elias Ghanem (Institut Jules Bordet, ULB, Belgium). All lines have been identified by RNA-seq and ATAC-seq. Mycoplasma test was performed systematically before each experiment that would provide data for this study.

All MM lines were cultured in Ham's F10 nutrient mix (Thermo Fisher Scientific, Waltham, MA) supplemented with 10% fetal bovine serum (Thermo Fisher Scientific) and 50 µg ml$^{-1}$ penicillin/streptomycin (Thermo Fisher Scientific). Cell cultures were kept at 37°C, with 5% $CO_2$.

## MPRA

The MPRA libraries were electroporated in 4–6 million cells each using the Nucleofector 2b or 4D (Lonza, Basel, Switzerland) with 6 µg of plasmid DNA and program T-030 or Y-001 and DS-132, EH-116, or CM-134, respectively. For the CHEQ-seq 5' and intron H3K27ac and CHEQ-seq ATAC libraries, one replicate was performed per cell line except for MM087, where three replicates were performed. For STARR-seq ATAC and the CHEQ-seq SOX-MITF combination library, one replicate was performed per cell line. For the CHEQ-seq enhancer tiling libraries, two replicates were performed per cell line except for MM087, where four replicates were performed. For the CHEQ-seq SOX10-KD library, three replicates were performed in MM087. Medium was changed 24 hr after electroporation. 48 hr post-electroporation, cells were detached from the plate using trypsin (Thermo Fisher Scientific). One-fifth of the cells was used for plasmid DNA extraction (QIAGEN, Hilden, Germany). The remaining cells underwent RNA extraction using the innuPREP RNA Mini Kit 2.0 (Analytik Jena, Jena, Germany), followed by mRNA isolation using the Dynabeads mRNA purification kit (Ambion, Austin, TX) and cDNA synthesis using the GoScript RT kit and oligo dT primer (Promega, Madison, WI). For STARR-seq samples, a junction PCR was done for 12 cycles with the primers 'STARR-seq_Junction_cDNA_For' or 'STARR-seq_Junction_plasmid_For' and 'STARR-seq_Junction_Rev' followed by a PCR to amplify the enhancer for four cycles with the primers 'STARR-seq_enhancer_Stag = X_For' and 'STARR-seq_enhancer_Rev.' For CHEQ-seq samples, a PCR was performed to amplify the random BC from the plasmid DNA or cDNA samples for 16 cycles with the primer pairs 'Cheq-seq_barcode_Intron_round1_Stag = 0_For'/'Cheq-seq_barcode_Intron_round1_Rev' for the CHEQ-seq intron H3K27ac library or 'Cheq-seq_barcode_5'_round1_Stag = X_For'/'Cheq-seq_barcode_5'_round1_Rev' for all other libraries. To add Illumina sequencing adaptors, all samples were finally amplified by PCR for six cycles with the primers 'i5_Indexing_For' and 'i7_Indexing_Rev.' After confirmation of the fragment size with a Bioanalyzer, samples were sequenced at the Genomics Core Leuven (KU Leuven).

## MPRA analysis

### Read processing and BC identification

Read processing following sequencing was performed with a custom bash script. First, random BCs were extracted using Cutadapt and filtered for a quality (Q-score) > 30. The number of reads per uniquely identified BC was counted and the name of the enhancer was assigned to the BC sequence based on the enhancer–BC assignment list for each library. Unassigned BCs were filtered out to obtain a final data frame containing the name of the enhancer, the BC sequence, and the number of reads.

### Estimation of enhancer activity

Enhancer activity from MPRA was estimated via a custom R script (RStudio, R version 3.6.0) (*Mauduit, 2021a*; https://github.com/aertslab/Melanoma_MPRA_paper; *Mauduit, 2021b*, copy archived at swh:1:rev:a90ee6e58d16fbaf367e4e68dc3a3d6fd3bc9e85). Enhancers were first filtered based on the number of BCs identified in the sequencing reads. Thresholds of 5 (for the H3K37ac libraries and enhancer tiling libraries), 10 (for ATAC libraries), or 20 (for the SOX-MITF combinations library) BCs per enhancer were selected based on the complexity of the library and the sequencing saturation of the enhancer–BC assignment samples. For the remaining enhancers, BC counts were aggregated per enhancer and then a count per million (CPM) normalization was applied. Plasmid (input) and cDNA (output) samples were merged by keeping only enhancers remaining in both samples after filtering. Input normalization was done by dividing CPM normalized cDNA counts by CPM normalized plasmid counts, resulting in an FC value. For libraries with shuffled sequences, a basal expression normalization was further applied by dividing the FC value of the enhancer by the median FC value of the shuffled sequences. The MPRAnalyze method was tried as an alternative to our method and gave nearly identical results in the case of the H3K27ac CHEQ-seq intron library (mean Pearson's correlation on log2 FC $r = 0.96$; *Ashuach et al., 2019*). The high computational demand when the number of BCs is high made it inappropriate for the analysis of most libraries. For consistency, all assays were analyzed with our aggregated BC method, which showed more consistency with low complexity libraries and more scalability with very complex libraries. In order to distinguish active and inactive enhancers, a Gaussian fit of the shuffled negative control values was performed with the 'robustbase 0.93-6' package, and a p-value and Benjamini–Hochberg adjusted p-value was calculated based on that Gaussian fit for all enhancers with the 'stats 3.6.0' package. An enhancer is considered active if its adjusted p-value

is <0.05. For the H3K27ac and ATAC-seq libraries, which did not contain shuffled sequences, regions containing no active tiles in the enhancer tiling MPRAs and displaying low activity in both H3K27ac and ATAC-seq libraries MPRAs were selected as negative controls and used to fit the Gaussian curve. For the CHEQ-seq SOX10-KD library, DEseq2 (*Love et al., 2014*) was used for estimating enhancer activity.

## Sample exclusion

Despite the high number of identified enhancer–random barcode couples during the assignment step for the CHEQ-seq enhancer tiling libraries A and B, the complexity of those libraries was so high that less than 3% of the barcodes could be identified following MPRA. This resulted in a low enhancer coverage or an insufficient number of remaining reads to identify enhancer activity in many samples. The 'OutlierD 1.48.0' R package was used to identify outliers. Samples that displayed <1% of outliers or had too low coverage (<450 tiles) were excluded.

## Motif enrichment and GSEA

Differential motif enrichment between the active tiles and the remaining tiles for the enhancer tiling libraries and the SOX10-KD library was performed via HOMER findMotifs (*Heinz et al., 2010*). For the enhancer tiling libraries, differential motif enrichment was also performed between the top 1/4th accessible tiles and the remaining tiles. MITF, SOX10, and TFAP2A ChIP-seq peaks (*Laurette et al., 2015*; *Seberg et al., 2017*) were intersected with the tiles using BEDtools. A GSEA was performed using the R package 'fgsea' (*Korotkevich et al., 2021*) by ranking the tiles according to their log2 FC and providing the overlaps of the ChIP-seq peaks with the tiles as gene sets.

## Deep learning predictions and nucleotide contribution visualization

Enhancer sequences for the enhancer tiling and SOX-MITF combinations libraries were scored with the DeepMEL2+ GABPA version of DeepMEL2 as described previously (*Atak et al., 2021*). To accommodate for the 500 bp required length of the sequences to be scored by DeepMEL2, the vector sequence flanking the insertion site of the enhancer was added on both sides of the sequence. A threshold of 0.1 was defined to distinguish between low and high prediction score for topics 16, 17, and 19 as it approaches the mean score +2 * standard deviation of those topics.

Visualization of nucleotide contribution to DeepMEL2 prediction score was done with DeepExplainer as described previously (*Atak et al., 2021*; *Lundberg and Lee, 2017*).

## Validation of TFBS functionality with luciferase assay

Three MES-specific ATAC regions (ABCC3_11-I, COL5A1_–17-D, GPR39_23-I), three MEL-specific ATAC regions (MLANA_5-I, IRF4_4-I, TYR_–9-D), and two inactive ATAC regions (SOX10_15–3UA, SOX10_–56DA) were selected. For each sequence, TFBSs for AP-1, TEAD, MITF, SOX, and ZEB were manually identified from the DeepExplainer profile and mutated the same way as for the enhancer tiling library. For ZEB binding sites, the CAGGTG site was mutated into CGGATG. A total of 24 sequences were ordered from Twist Bioscience pre-cloned in the pTwist ENTR plasmid (*Supplementary file 1*). Regions were introduced in the pGL4.23-GW luciferase reporter plasmid (Promega) via Gateway LR recombination reaction (Invitrogen) and transformed into Stellar chemically competent bacteria (Takara). Plasmid minipreps were performed using the NucleoSpin Plasmid Transfection-grade Mini kit (Macherey-Nagel) and sequenced with Sanger sequencing to confirm the correct insertion of the regions in the destination plasmid.

MM001 and MM099 were seeded in 24-well plates and transfected with 400 ng pGL4.23-Enhancer plasmid + 40 ng pRL-TK renilla plasmid (Promega) with Lipofectamine 2000. One day after transfection, luciferase activity was measured via the Dual-Luciferase Reporter Assay System (Promega) by following the manufacturer's protocol. Briefly, cells were lysed with 100 µl of Passive Lysis Buffer for 15 min at 500 rpm. 20 µl of the lysate were transferred in duplicate in a well of an OptiPlate-96 HB (PerkinElmer, Waltham, MA) and 100 µl of Luciferase Assay Reagent II were added in each well. Luciferase-generated luminescence was measured on a Victor X luminometer (PerkinElmer). 100 µl of the Stop & Glo Reagent was added to each well, and the luminescence was measured again in order to record renilla activity. Luciferase activity was estimated by calculating the ratio luciferase/renilla. This value was further normalized by the ratio calculated on blank wells containing only reagents. Three

biological replicates were done per condition, except for all MLANA_5-I and IRF4_4-I sequences in MM001, where four biological replicates were done.

## Transcription factor ChIP-seq analysis

ChIP-seq data for MITF, SOX10, JUN, and JUNB were CPM normalized with the bamCoverage function (deepTools; v3.3.1) (*Ramírez et al., 2016*). For analysis of signal over the H3K27ac regions selected for the CHEQ-seq assay, the mean signal over the whole region was calculated with the 'bigWigAverageOverBed' function (kentUtils). For the heatmaps of the TFs ChIP-seq signal over the differentially acetylated MEL and MES regions from *Verfaillie et al., 2015*, the 'computeMatrix scale-regions' function followed by the 'plotHeatmap' function (deepTools; v3.3.1) were used. To perform a t-test on TFs ChIP-seq signal in MEL and MES regions, reads per region were first counted with 'featureCount' from the 'Rsubread' R package (*Liao et al., 2019*).

## ChIP-seq, ATAC-seq, and RNA-seq public data

### Publicly available data used in this work

MITF and SOX10 ChIP-seq in 501mel were downloaded from the GEO entry GSE61965 (*Laurette et al., 2015*). TFAP2A ChIP-seq in human primary melanocytes purified from discarded neonatal foreskin samples was downloaded from the GEO entry GSE67555 (*Seberg et al., 2017*). JUN and JUNB ChIP-seq in MM099 line were downloaded from the GEO entry GSE159965 (*Atak et al., 2021*). H3K27ac ChIP-seq for MM001, MM011, MM031, MM034, MM047, MM057, MM074, MM087, MM099, MM118, and SKMEL5 were downloaded from the GEO entries GSE60666 and GSE114557 (*Bravo González-Blas et al., 2019*; *Verfaillie et al., 2015*). OmniATAC-seq data for MM001, MM011, MM029, MM031, MM034, MM047, MM057, MM074, MM087, MM099, and MM118 were downloaded from the GEO entries GSE142238 and GSE134432 (*Minnoye et al., 2020*; *Wouters et al., 2020*). SOX10-KD time-course OmniATAC-seq for MM057 and MM087 were downloaded from the GEO entry GSE114557 (*Bravo González-Blas et al., 2019*). Single-cell RNA-seq data for MM001, MM029, MM047, MM057, MM074, MM087, and MM099 were downloaded from the GEO entry GSE134432 (*Wouters et al., 2020*).

### H3K27ac ChIP-seq processing

H3K27ac ChIP-seq data used in this study were processed in a previous study (*Verfaillie et al., 2015*) as follows. Briefly, ChIP-seq reads were mapped to the genome (hg19 with GENCODE v18 annotation) using Bowtie 2 v2.1.0 (*Langmead and Salzberg, 2012*). The coverage of candidate regulatory regions was computed using BEDtools. Subsequently, read counts were transformed using a regularized log transformation. The DESeq function from R/Bioconductor package DESeq2 v1.4.5 was used to detect differentially active regions between the two invasive and nine proliferative samples. p-Values were adjusted for multiple testing according to Benjamin–Hochberg. On the H3K27ac signal, applying a threshold adjP ≤ 0.05 and log2FC ≥|1| led to 13,671 regions that were more active in invasive samples and 7146 regions more active in proliferative samples. This strategy was then compared with differential peak calling by MACS2 algorithm (q < 0.05, nomodel), with the proliferative samples as treatment and invasive samples as control. This supported the differentially called regions, resulting in final sets of 13,453 invasive and 6,669 proliferative regions.

### ATAC-seq processing

ATAC-seq data used in this study were processed in a previous study (*Minnoye et al., 2020*) as follows. Briefly, reads were mapped to the human genome (hg19-GENCODE v18) using Bowtie 2 v2.2.6 (*Langmead and Salzberg, 2012*). Mapped reads were sorted using SAMtools v1.8 (*Li et al., 2009*), and duplicates were removed using Picard MarkDuplicates v1.134. Reads were filtered by removing mitochondrial reads and filtering for Q > 30 using SAMtools. Peaks were called using MACS2 v2.1.2 (*Gaspar, 2018*). Blacklisted regions (ENCODE) and peaks overlapping with alternative chromosomes and ChrM were removed. Summits were extended by 250 bp up- and downstream using slopBed (BEDtools; v2.28.0) (*Quinlan and Hall, 2010*), providing human chromosome sizes. Peaks were normalized for the library size and overlapping peaks were filtered using the peak score by keeping the peak with the highest score. Normalized bigWigs were either made from normalized

bedGraphs using as scaling parameter (-scale) $1 \times 10^6$/(number of non-mitochondrial mapping reads); or made by bamCoverage (deepTools; v3.3.1) (*Ramírez et al., 2016*).

## Data access

MPRA data generated for this study have been submitted to the NCBI Gene Expression Omnibus (GEO, https://www.ncbi.nlm.nih.gov/geo/) under accession number GSE180879.

## Code availability

We have made the scripts for the most important analyses, including enhancer tiling design, enhancer–barcode assignment, reads processing, MPRA analysis, HOMER analysis, and ChIP-seq and ATAC-seq analysis available from the laboratory's GitHub website (https://github.com/aertslab/Melanoma_MPRA_paper; *Mauduit, 2021b*).

## Acknowledgements

This work was supported by an ERC Consolidator Grant to SA (no. 724226_cis-CONTROL), the KU Leuven (grant no. C14/18/092 to SA), the Foundation Against Cancer (grant no, 2016-070 to SA), a PhD and a postdoctoral fellowship from the FWO (LM, no. 1S03317N, JD no. 12J6916N, respectively) and a postdoctoral research fellowship from Kom op tegen Kanker (Stand up to Cancer), the Flemish Cancer Society, and from Stichting tegen Kanker (Foundation against Cancer), the Belgian Cancer Society (JW). Computing was performed at the Vlaams Supercomputer Center and high-throughput sequencing via the Genomics Core Leuven. MM lines were a kind gift from Pr. Ghanem-Elias Ghanem (Institut Jules Bordet, ULB, Belgium). The funders had no role in study design, data collection and analysis, decision to publish, or preparation of the manuscript.

## Additional information

### Funding

| Funder | Grant reference number | Author |
|---|---|---|
| H2020 European Research Council | 724226_cis-CONTROL | Stein Aerts David Mauduit Valerie Christiaens |
| KU Leuven | C14/18/092 | Stein Aerts |
| Fonds Wetenschappelijk Onderzoek | 1S03317N | Liesbeth Minnoye |
| Fonds Wetenschappelijk Onderzoek | 12J6916N | Jonas Demeulemeester |
| Kom op tegen Kanker | | Jasper Wouters |
| Stichting tegen Kanker | 2019-100 | Jasper Wouters |

The funders had no role in study design, data collection and interpretation, or the decision to submit the work for publication.

### Author contributions

David Mauduit, Liesbeth Minnoye, Conceptualization, Data curation, Formal analysis, Investigation, Methodology, Validation, Visualization, Writing – original draft; Ibrahim Ihsan Taskiran, Maxime de Waegeneer, Formal analysis, Investigation, Software, Visualization; Valerie Christiaens, Data curation, Methodology, Project administration; Gert Hulselmans, Formal analysis, Resources, Software; Jonas Demeulemeester, Formal analysis, Investigation, Writing – original draft; Jasper Wouters, Formal analysis, Writing – original draft; Stein Aerts, Conceptualization, Funding acquisition, Investigation, Methodology, Project administration, Supervision, Writing – original draft

### Author ORCIDs

David Mauduit  http://orcid.org/0000-0002-2045-227X

Ibrahim Ihsan Taskiran 🆔 http://orcid.org/0000-0002-5077-5264
Jonas Demeulemeester 🆔 http://orcid.org/0000-0002-2660-2478
Jasper Wouters 🆔 http://orcid.org/0000-0002-7129-2990
Stein Aerts 🆔 http://orcid.org/0000-0002-8006-0315

## Decision letter and Author response

Decision letter https://doi.org/10.7554/eLife.71735.sa1
Author response https://doi.org/10.7554/eLife.71735.sa2

## Additional files

### Supplementary files

• Supplementary file 1. Composition of the massively parallel reporter assay (MPRA) libraries and list of sequences tested by luciferase assay. For the ATAC-seq-based library, modifications to the original sequence to allow DNA synthesis are indicated in the 'Modification' column. For the SOX-MITF combinations library, a table indicating the arrangement of the binding sites is provided.

• Supplementary file 2. List of primers used for the generation of each massively parallel reporter assay (MPRA) library and for the sequencing library preparation.

• Transparent reporting form

### Data availability

Sequencing data have been deposited in GEO under accession codes GSE180879. Enhancer activity tables for each library is provided as source data. Scripts used for enhancer - barcode assignment, read processing and activity measurement and analysis are provided in the Scripts directory.

The following dataset was generated:

| Author(s) | Year | Dataset title | Dataset URL | Database and Identifier |
|---|---|---|---|---|
| Mauduit D, Minnoye L, Taskiran II M, Christiaens V, Hulselmans G, Demeulemeester J, Wouters J, Aerts S | 2021 | Analysis of long and short enhancers in melanoma cell states | https://www.ncbi.nlm.nih.gov/geo/query/acc.cgi?acc=GSE180879 | NCBI Gene Expression Omnibus, GSE180879 |

The following previously published datasets were used:

| Author(s) | Year | Dataset title | Dataset URL | Database and Identifier |
|---|---|---|---|---|
| Laurette P, Davidson I | 2015 | BRG1 recruitment by transcription factors MITF and SOX10 defines a specific configuration of regulatory elements in the melanocyte lineage (ChIP-seq) | https://www.ncbi.nlm.nih.gov/geo/query/acc.cgi?acc=GSE61965 | NCBI Gene Expression Omnibus, GSE61965 |
| Van Otterloo E, Cornell RA, Seberg HE | 2017 | TFAP2A ChIP-seq in human primary melanocytes | https://www.ncbi.nlm.nih.gov/geo/query/acc.cgi?acc=GSE67555 | NCBI Gene Expression Omnibus, GSE67555 |
| Kalender-Atak Z, Ihsan-Taskiran I, Wouters J, Hulselmans G, Aerts S | 2021 | Prioritization of enhancer mutations by combining allele-specific chromatin accessibility with motif analysis and deep learning | https://www.ncbi.nlm.nih.gov/geo/query/acc.cgi?acc=GSE159965 | NCBI Gene Expression Omnibus, GSE159965 |

*Continued on next page*

*Continued*

| Author(s) | Year | Dataset title | Dataset URL | Database and Identifier |
|---|---|---|---|---|
| Atak ZK | 2015 | Decoding the regulatory landscape of melanoma reveals TEADS as regulators of the invasive cell state | https://www.ncbi.nlm.nih.gov/geo/query/acc.cgi?acc=GSE60666 | NCBI Gene Expression Omnibus, GSE60666 |
| Bravo Gonzalez Blas C, Minnoye L, Aerts S | 2019 | cisTopic: cis-regulatory topic modelling on single-cell ATAC-seq data | https://www.ncbi.nlm.nih.gov/geo/query/acc.cgi?acc=GSE114557 | NCBI Gene Expression Omnibus, GSE114557 |
| Minnoye L, Taskiran II AS, Zon L, Yang S | 2020 | Cross-species analysis of melanoma enhancer logic using deep learning | https://www.ncbi.nlm.nih.gov/geo/query/acc.cgi?acc=GSE142238 | NCBI Gene Expression Omnibus, GSE142238 |
| Wouters J, Kalender-Atak Z, Christiaens V, Spanier KI, Aerts S | 2020 | Single-cell analysis of gene expression variation and phenotype switching in melanoma | https://www.ncbi.nlm.nih.gov/geo/query/acc.cgi?acc=GSE134432 | NCBI Gene Expression Omnibus, GSE134432 |

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
