## [Editor Report]

This study describes an integrative analysis of the location, regulation, and function of melanoma cell state-specific enhancer elements. By comparing enhancer activity through massively parallel reporter assays, chromatin features, and underlying TF binding profiles in melanocytic and mesenchymal-like melanoma cell states, the authors identify candidate regulators and mechanisms that explain enhancer activity and specificity in melanoma biology. These findings will be of broad interest to those seeking to understand cell type- or cell identify-specific gene regulation at the level of transcriptional and epigenetic control of *cis*-regulatory elements.

---

## [Decision Letter]

**Decision letter after peer review:**

Thank you for submitting your article "Analysis of long and short enhancers in melanoma cell states" for consideration by *eLife*. Your article has been reviewed by 3 peer reviewers, including Jian Xu as Reviewing Editor and Reviewer #1, and the evaluation has been overseen by Richard White as the Senior Editor. The reviewers have opted to remain anonymous.

Essential revisions:

1) Provide more details on the filtering and selection criteria for the identification of MES- and MEL-specific regions. A simple flowchart to illustrate the major selection criteria might be helpful.

2) Consider improving the nomenclature for various enhancer regions, as commented by Reviewer #1.

3) For the identification of candidate TFs and their combinations responsible for differential enhancer activity in MEL vs MES cells, DeepMEL2-based analysis does not seem to provide robust predictions. This is also exemplified by the analysis of binding sites for MITF and AP-1, as commented by Reviewer #2. Therefore, the authors may use available ChIP-seq data for candidate TFs and quantitatively compare signals at the respective enhancers in different cell states. This analysis may be further extended to other differentially enriched H3K27ac and/or ATAC-seq peaks to evaluate whether the identified TFs are generally associated with differential H3K27ac and/or ATAC-seq signals between cell states.

4) Provide information about AP-1 expression in MEL-intermediate cells with and without SOX10 knockdown, as commented by Reviewer #2.

5) While altering TF binding sites at native gene loci may be technically challenging (and would be good for future studies), it would be helpful to include an analysis of SOX10, MITF or AP-1 mutations in the context of a larger MPRA sequence on a few test cases to determine the effect on enhancer activity, as commented by Reviewer #3. This would be informative for future investigators that the MPRA assay would be a useful method for figuring out which TF sites would be good to go after in vivo.

6) Additional discussion on the functional roles of SOX10 and MITF in melanoma pathophysiology would be helpful to highlight the broad relevance of findings in the current study.

7) Consider revising figures to increase readability.

*Reviewer #1 (Recommendations for the authors):*

1. It would be helpful to provide more details on the filtering and selection criteria for the identification of 35 MES- and 18 MEL-specific regions as H3K27ac ChIP-seq based library. Is there any normalization steps for ChIP-seq and ATAC-seq signals across samples? How are enhancer target genes defined? A simple flowchart to illustrate the major selection criteria might be helpful.

2. Line 102, 18 MES-specific ATAC-seq peaks were selected from 35 MES-specific H3K27ac peaks. Are there H3K27ac peaks without selected ATAC-seq peaks? If so, why were those H3K27ac peaks excluded for the current study?

3. It is advised to improve the nomenclature for various enhancer regions. Instead of using Gene_1, Gene_2, etc, it would be more informative to show the location and distance of the enhancers to the TSS of putative gene targets. For example, Gene_+10kb indicates an enhancer located 10kb upstream of the TSS while Gene_-10kb indicates 10kb downstream of TSS. The distance could be determined by distance between the peak summit of ATAC-seq (or H3K27ac if they overlap) and TSS.

4. For the identification of candidate TFs and their combinations responsible for differential enhancer activity in MEL vs MES cells, DeepMEL2-based analysis does not seem to provide robust predictions. Therefore, the authors may also use available ChIP-seq data for candidate TFs and quantitatively compare the normalized ChIP-seq signals at the respective enhancers in different cell states. This analysis may also be extended to include other differentially enriched H3K27ac and/or ATAC-seq peaks to evaluate whether the identified TFs (e.g. SOX10-MITF for MEL and AP1 for MES) are generally associated with differential H3K27ac and/or ATAC-seq signals between cell states.

*Reviewer #2 (Recommendations for the authors):*

This manuscript is mostly convincing, but a few aspects of data need to be clarified to strengthen the main conclusions.

1. In Figure 1a, a MITF binding site was predicted within IRF4 enhancer and a serial of tiles with mutated MITF motif were constructed based on the prediction. However, in comparison with the MITF ChIP-seq track, the predicted MITF motif apparently located outside of the MITF ChIP peak. This is also true for predicted AP-1 binding sites in COL5A1 and HEG1 enhancers. How accurately can those predicted TF bind sites based on DNA sequence represent the TF binding on chromatin in vivo?

2. SOX10 KD shifts MEL-intermediate cell to MES phenotype and shapes the landscape of chromatin accessibility. Upon SOX10 KD, does AP-1 expression increase and gained ATAC-seq peaks display high predictions score for topic 19?

3. Authors conclude that AP-1 binding alone drives MES state specific enhancer activity. However, MES enhancers, such as COL5A1_5 region, show strong enhancer activities by reporter assay in MEL-intermediate cell as well, despite that the endogenous loci remain close in those cells (as shown in Figure 6a). How much AP-1 protein is expressed in MEL-intermediate cells?

*Reviewer #3 (Recommendations for the authors):*

To address weaknesses:

1) To better contextualize the role of specific TFs in regulating a given gene in Weakness 1 – Altering or deleting TF binding sites or subregions of specific enhancers, especially at the scale of test sequences discussed in this study, at the native gene locus seems destined for future studies, although could be argued is the truest measure of the role of an individual TF site/enhancer subregion. As a potentially more intermediate/approachable assay, could an analysis of SOX or MITF or AP-1 mutations in the context of a larger MPRA sequence (e.g. 510 bp or in the H3K27Ac 1-2 kb sequence) in addition to the tiled versions (Supp Figure 7 or 10) be done on a few test cases to capture enhancer context better. As the authors note on page 19, lines 496-497, ~500 bp sequences may be needed to capture sufficient complexity of the enhancer.

2) Brief mention of the demonstrated functional role of SOX10 and MITF protein in melanoma onset/growth/survival could be included (e.g. shRNA knockdown of sox10 decreasing human melanoma growth in culture or knockdown/deletion delaying/blocking melanoma in animal models). The binding sites for these TFs are clearly relevant in the assays shown, but the larger context/support for these TFs role in melanomagenesis could thus be better highlighted. Similarly, SOXE dimer sites have been noted to be over-represented in melanoma-specific DNase I hypersensitive sites previously (Huang et al., and Jauch, Scientific Reports, 2015) and supports the papers conclusions.

Data availability appears appropriate.

---

## [Author Response]

Reviewer #1 (Recommendations for the authors):1. It would be helpful to provide more details on the filtering and selection criteria for the identification of 35 MES- and 18 MEL-specific regions as H3K27ac ChIP-seq based library. Is there any normalization steps for ChIP-seq and ATAC-seq signals across samples?

We apologise for this omission and now provide the following details. H3K27ac ChIP-seq data used in this study were processed in a previous study (Verfaillie et al., 2015) as follows: ChIP-seq reads were mapped to the genome (hg19 with GENCODE v18 annotation) using Bowtie2 v2.1.0. The coverage of candidate regulatory regions was computed using BEDTools. Subsequently, read counts were transformed using a regularized log transformation. The *DESeq* function from R/Bioconductor package DESeq2 v1.4.5 was used to detect differentially active regions between the two invasive and nine proliferative samples. P-values were adjusted for multiple testing according to Benjamin-Hochberg. On the H3K27ac signal, applying a threshold adjP ≤0.05 and log2FC ≥|1| led to 13,671 regions that were more active in invasive samples and 7,146 regions more active in proliferative samples. This strategy was then compared with differential peak calling by MACS2 algorithm (*q*<0.05, *nomodel*), with the proliferative samples as treatment and invasive samples as control. This supported the differentially called regions resulting in final sets of 13,453 invasive and 6,669 proliferative regions.

Downstream analysis and filtering process resulting in the selection of the 35 MES- and 18 MEL-specific regions are described in the “Methods > MPRA design and cloning > H3K27ac based library” section and illustrated with a flowchart in Supplementary Figure 1.

We have limited ourselves to the selection of 65 H3K27ac regions (down to 53 after some amplification failures) because we had to design primers and perform PCR amplification from genomic DNA for each sequence individually.

ATAC-seq data used in this study were processed in a previous study (Minnoye et al., 2020) as follows: Reads were mapped to the human genome (hg19-GENCODE v18) using Bowtie 2 (v2.2.6) (Langmead and Salzberg 2012). Mapped reads were sorted using SAMtools (v1.8) (Li et al., 2009), and duplicates were removed using Picard MarkDuplicates (v1.134). Reads were filtered by removing mitochondrial reads and filtering for Q> 30 using SAMtools. Peaks were called using MACS2 (v2.1.2) (Gaspar 2018). Blacklisted regions (ENCODE) and peaks overlapping with alternative chromosomes and ChrM were removed. Summits were extended by 250 bp up- and downstream using slopBed (BEDTools; v2.28.0) (Quinlan and Hall 2010), providing human chromosome sizes. Peaks were normalized for the library size, and overlapping peaks were filtered using the peak score by keeping the peak with the highest score. Normalized bigWigs were either made from normalized bedGraphs using as scaling parameter (-scale) 1 × 10^6^/(number of non mitochondrial mapping reads); or made by bamCoverage (deepTools, v3.3.1) (Ramírez et al., 2016).

This extra information has been added to the “Methods > ChIP-seq, ATAC-seq and RNA-seq public data” section.

How are enhancer target genes defined?

The assignment of target genes to the selected regions was indeed not clearly mentioned. Following i-cisTarget analysis and gene filtering, genes are ordered by differential expression in MES vs MEL lines. The TSS of the top and bottom genes is used as the starting point to manually search for H3K27ac regions specific for MES and MEL respectively. Identified regions, within ~150kb of the TSS, are associated with the differentially expressed gene, even if some of them are closer to or within another gene.

An exception was made for *CDH1* enhancers. *CDH3* TSS was used as a starting point to search for H3K27ac regions. Two differentially acetylated regions were found in an intron of *CDH1* downstream of *CDH3* (Author response Figure 1). Because *CDH1* is also part of the differentially expressed genes used to search for regions, both selected regions have been assigned to *CDH1*. Both genes are more expressed in MEL lines so it is possible that those enhancers regulate them both.

**Author response image 1. sa2fig1:** Location of CDH1_1 and CDH1_2 regions in regard to CDH1 and CDH3 genes.

The part covering the manual selection of the H3K27ac in the Methods section has been rephrased.

A simple flowchart to illustrate the major selection criteria might be helpful.

We thank the reviewer for this suggestion and have now added the flowchart as a novel panel in Figure 1—figure supplement 1 (and refer to it in the text).

2. Line 102, 18 MES-specific ATAC-seq peaks were selected from 35 MES-specific H3K27ac peaks. Are there H3K27ac peaks without selected ATAC-seq peaks? If so, why were those H3K27ac peaks excluded for the current study?

Out of 53 MES/MEL-specific H3K27ac regions, 37 had ATAC-seq peaks selected in them (for 9 regions 2 ATAC-seq peaks per region were selected). Of the 16 H3K27ac regions where no ATAC-seq peak was selected, one does not contain ATAC-seq peaks (*MAP3K14*_1) and the 15 others did not display activity in the H3K27ac MPRA assay and did not overlap with topic 4 (MEL-specific) or topic 7 (MES-specific) regulatory regions from our previously published cisTopic analysis of ATAC-seq data from 16 human melanoma cell lines (Bravo González-Blas et al., 2019).

Extra information clarifying this selection process has been added to the “Design of MPRA libraries based on H3K27ac, ATAC-seq, and synthetic sequences” part of the Results section.

3. It is advised to improve the nomenclature for various enhancer regions. Instead of using Gene_1, Gene_2, etc, it would be more informative to show the location and distance of the enhancers to the TSS of putative gene targets. For example, Gene_+10kb indicates an enhancer located 10kb upstream of the TSS while Gene_-10kb indicates 10kb downstream of TSS. The distance could be determined by distance between the peak summit of ATAC-seq (or H3K27ac if they overlap) and TSS.

We thank the reviewer for this suggestion. We changed the name format of the enhancer regions for the H3K27ac library to GENE_(-)00-X where “(-)00” is the distance (in kb) between the TSS and the highest peak summit of the H3K27ac region. “X” is the gene annotation of the region with “P” for promoter, “I” for Intron, “E” for Exon, “D” for distal and “3U” for 3’ UTR. This annotation is based on the GENCODE V38 (GRCh37) basic gene annotation. For example, TYR_-9-D (previously TYR_1) in a distal region located 9 kb upstream of the *TYR* TSS.

This nomenclature is conserved for the ATAC based library (e.g. TYR_-9-D keeps the same name in both the H3K27ac library and the ATAC library). When 2 ATAC regions are located within the same H3K27ac region, they are distinguished by suffixes A and B (e.g. KIT_1A and KIT_1B are now KIT_114-DA and KIT_114-DB).

A description of this new nomenclature has been added to the method section of the paper.

4. For the identification of candidate TFs and their combinations responsible for differential enhancer activity in MEL vs MES cells, DeepMEL2-based analysis does not seem to provide robust predictions. Therefore, the authors may also use available ChIP-seq data for candidate TFs and quantitatively compare the normalized ChIP-seq signals at the respective enhancers in different cell states. This analysis may also be extended to include other differentially enriched H3K27ac and/or ATAC-seq peaks to evaluate whether the identified TFs (e.g. SOX10-MITF for MEL and AP1 for MES) are generally associated with differential H3K27ac and/or ATAC-seq signals between cell states.

Thank you for the suggestion. We used ChIP-seq data for MITF, SOX10 and AP-1 (JUN and JUNB) generated in melanoma cell lines and compared them to the CHEQ-seq values of the H3K27ac regions tested in this study. As expected, regions with high MITF and/or SOX10 ChIP-seq also have high CHEQ-seq activity in MM001 (MEL) and regions with high AP-1 ChIP-seq also have high CHEQ-seq activity in MM029 (MES). Extending the analysis to all MEL vs MES differentially acetylated regions from Verfaillie et al., 2015, we find a significant enrichment of MITF/SOX10 and AP-1 ChIP-seq signal in MEL and MES regions respectively (t-test p-value < 2.2e-16 for each TF).

These results have been added to the relevant Results section of the paper and in Figure 5—figure supplement 2 and Figure 5—figure supplement 3.

Reviewer #2 (Recommendations for the authors):This manuscript is mostly convincing, but a few aspects of data need to be clarified to strengthen the main conclusions.1. In Figure 1a, a MITF binding site was predicted within IRF4 enhancer and a serial of tiles with mutated MITF motif were constructed based on the prediction. However, in comparison with the MITF ChIP-seq track, the predicted MITF motif apparently located outside of the MITF ChIP peak. This is also true for predicted AP-1 binding sites in COL5A1 and HEG1 enhancers. How accurately can those predicted TF bind sites based on DNA sequence represent the TF binding on chromatin in vivo?

We decided to mutate only the highly conserved motifs in order to keep the number of tiles manageable (Author response image 2). Unfortunately, this resulted in some functional sites not being recognised by PWM scores. In the case of the IRF4__4-I region, as mentioned by the reviewer, there is one MITF motif identified by its PWM that does not overlap with a ChIP-seq peak, and one ChIP-seq peak without a high-scoring PWM match. Interestingly, the DeepExplainer profile of the tile containing the high-scoring PWM match (Author response image 2) shows the presence of neighboring ZEB repressor motifs, potentially explaining the absence of MITF ChIP-seq signal at this location. The DeepExplainer profile of the IRF4__4-I ATAC-seq region (Author response image 2) shows a MITF motif with a predicted contribution to the ATAC-seq signal and which overlaps with the MITF ChIP-seq peak. A mutation in nucleotide 10 of the motif (A instead of C or T) is most likely the reason for this motif not being picked up as PWM match even though the ChIP-seq data suggest that it is functional. It is interesting to see that DeepMEL2 allows more flexibility in the identification of functional TFBSs by taking motif context into consideration, as illustrated in figure 6 for the COL5A1_-17-D region. Regarding the alignment of SOX and AP-1 motifs, Author response image 2. helps to see that they correctly overlap with the corresponding ChIP-seq peaks.

Details regarding the mutation of only highly conserved motifs has been added to the “Methods > MPRA design and cloning > Enhancer tiling libraries” section.

**Author response image 2. sa2fig2:** (**a**) Sequence logo visualisation of the MITF motif position weight matrix. (**b**) DeepExplainer profile of the tile “chr6:395485-395675@@IRF4_4-I==wt” containing the MITF motif identified by ClusterBuster using the MITF PWM (green highlight) and flanking ZEB motifs (grey highlights). (**c**) DeepExplainer profile of the IRF4_4-I ATAC-seq region. The MITF motif (not identified by ClusterBuster) is highlighted in green. (**d**) Tracks showing ChIP-seq signal, location of selected motifs and tiles. Grey vertical lines highlight the overlap of the selected motif with ChIP-seq peak summit for SOX10 and AP-1*.*

2. SOX10 KD shifts MEL-intermediate cell to MES phenotype and shapes the landscape of chromatin accessibility. Upon SOX10 KD, does AP-1 expression increase and gained ATAC-seq peaks display high predictions score for topic 19?

Data from Wouters et al., 2020 show that upon *SOX10* knock down (KD), several members of the AP-1 family are upregulated in MM087 (Author response image 3). The same is observed for MM057 and MM074. At the chromatin accessibility level, SOX10 KD results in increased accessibility of topic 19 regions (Author response image 3). Mention of AP-1 expression and increased chromatin accessibility following SOX10 KD in intermediate MEL cell lines has been added to the first paragraph of the “Results > MES enhancers show lower but consistent activity in intermediate lines” section with Wouter et al., 2020 as reference.

**Author response image 3. sa2fig3:** (**a**) Transcript expression level of AP-1 members upon SOX10 knock down in MM087 at different time points. BL: Base Line, NTC: Non-Targeting Control. (**b**) Heatmap of MM087 ATAC-seq signal in topic 19 regions upon SOX10 knock down*.*

3. Authors conclude that AP-1 binding alone drives MES state specific enhancer activity. However, MES enhancers, such as COL5A1_5 region, show strong enhancer activities by reporter assay in MEL-intermediate cell as well, despite that the endogenous loci remain close in those cells (as shown in Figure 6a). How much AP-1 protein is expressed in MEL-intermediate cells?

We do not have AP-1 protein expression data for MEL-intermediate lines. However, transcript levels from scRNA-seq data (Wouters et al., 2020) show expression of FOSL1 in all MEL lines except MM001 (in Wouters et al., we call these MEL lines intermediate or transitory after Tsoi et al.,), as well as in MES lines (Author response image 4). If we look at the track-based FOSL1 regulon activity, we also see a gradient between MM001, intermediate MEL and MES lines (Author response image 4) further confirming AP-1 activity. Mention of AP-1 expression in MEL-intermediate cells has been added to the “Results > AP-1 sites alone can explain MES enhancer activity” section.

**Author response image 4. sa2fig4:** (**a**) Transcript expression level of AP-1 members from scRNA-seq of melanoma cell lines. (**b**) Track-based FOSL1 regulon activity.

Reviewer #3 (Recommendations for the authors):To address weaknesses:1) To better contextualize the role of specific TFs in regulating a given gene in Weakness 1 – Altering or deleting TF binding sites or subregions of specific enhancers, especially at the scale of test sequences discussed in this study, at the native gene locus seems destined for future studies, although could be argued is the truest measure of the role of an individual TF site/enhancer subregion. As a potentially more intermediate/approachable assay, could an analysis of SOX or MITF or AP-1 mutations in the context of a larger MPRA sequence (e.g. 510 bp or in the H3K27Ac 1-2 kb sequence) in addition to the tiled versions (Supp Figure 7 or 10) be done on a few test cases to capture enhancer context better. As the authors note on page 19, lines 496-497, ~500 bp sequences may be needed to capture sufficient complexity of the enhancer.

We thank the reviewer for their suggestions. To confirm the function of the SOX, MITF and AP-1 in a larger context, we now generated mutant versions of 501 bp ATAC regions and evaluated their activity via luciferase assay in both a MEL and a MES line.

We used the ATAC regions instead of H3K27ac regions for the following reasons:

1. As opposed to H3K27ac regions (which vary between 1.2 and 2.9 kb), all ATAC region sequences are 501 bp long, resulting in luciferase plasmids with identical size to avoid measurement bias due to variability in transfection efficiency.

2. The ATAC regions correspond to the accessible part of the genome where the TFs are bound.

3. The ATAC regions recapitulate the enhancer activity of the broader H3K27Ac regions.

4. With lengths up to several kb, as is the case for many H3K27ac regions, synthesis becomes impractical.

Validation via luciferase assay instead of MPRA allows us to simultaneously study the effect of TFBS mutation in the context of ATAC regions and validate our MPRA results in a different assay, by re-assessing the activity of wild type sequences.

Three MES-specific (ABCC3_11-I, COL5A1_-17-D, GPR39_23-I), three MEL-specific (MLANA_5-I, IRF4_4-I, TYR_-9-D) and two inactive ATAC regions (SOX10_15-3UA, SOX10_-56-DA) were selected. For each sequence, TFBSs for AP-1, MITF, SOX and ZEB were manually identified from the DeepExplainer (DeepMEL2) profiles and mutated the same way as for the enhancer tiling library. For ZEB binding sites, the CAGGTG site was mutated into CGGATG (mutation of the 2 nucleotides with the highest weight in the hocomoco__ZEB1_HUMAN.H10MO.B PWM). A total of 24 sequences were ordered from TWIST Bioscience pre-cloned in the pTwist ENTR plasmid (Supplementary table 1). Regions were introduced in the pGL4.23-GW luciferase reporter plasmid (Promega) via Gateway LR recombination reaction (Invitrogen) and transformed into Stellar chemically competent bacteria (Takara). Plasmid minipreps are performed using the NucleoSpin Plasmid Transfection-grade Mini kit (Macherey-Nagel) and are sequenced with Sanger sequencing to confirm the correct insertion of the regions in the destination plasmid.

We opted to use MM001 (MEL) and MM099 (MES) for these experiments. MM099 was selected over MM029 because of its better transfectability with Lipofectamine 2000.

The luciferase assay confirmed both the activity and specificity of the wild type sequences in the target cell type and the importance of each TF for enhancer activity. A significant drop in luciferase activity is observed in MM001 and MM099 when, respectively, MITF/SOX and AP-1 predicted binding sites are mutated. The mutation of ZEB motifs results in an increase in activity of MLANA_5-I and IRF4_4-I in MM001, as predicted by DeepMEL2, whereas in MES enhancer the effect of ZEB is found to be weaker.

Results and Methods have been added to the paper and the figure has been added as Figure 5—figure supplement 3 and Figure 6—figure supplement 3.

2) Brief mention of the demonstrated functional role of SOX10 and MITF protein in melanoma onset/growth/survival could be included (e.g. shRNA knockdown of sox10 decreasing human melanoma growth in culture or knockdown/deletion delaying/blocking melanoma in animal models). The binding sites for these TFs are clearly relevant in the assays shown, but the larger context/support for these TFs role in melanomagenesis could thus be better highlighted. Similarly, SOXE dimer sites have been noted to be over-represented in melanoma-specific DNase I hypersensitive sites previously (Huang et al., and Jauch, Scientific Reports, 2015) and supports the papers conclusions.

The following text has been added to the “Result > Key transcription factor binding sites explain melanoma enhancer activity” section:

“SOX10 and MITF both represent established lineage transcription factors in melanocytes and in melanoma and TFAP2 has been found to co-occurrence with MITF in active regulatory elements in melanocytes (Goding, 2000; Harris et al., 2010; Seberg et al., 2017). SOX10 was previously shown to be necessary for melanoma formation and maintenance by controlling cell survival and cell cycle (Cronin et al., 2013; Shakhova et al., 2012). MITF is considered to be a melanoma oncogene (Garraway and Sellers, 2006) and has been implicated in various cellular processes (Goding and Arnheiter, 2019; Levy et al., 2006).”